# ELViS: Efficient Visual Similarity from Local Descriptors that Generalizes Across Domains

**Pavel Suma**[1]    **Giorgos Kordopatis-Zilos**[1]    **Yannis Kalantidis**[2]    **Giorgos Tolias**[1]

[1]VRG, FEE, Czech Technical University in Prague    [2]NAVER LABS Europe

## Abstract

Large-scale instance-level training data is scarce, so models are typically trained on domain-specific datasets. Yet in real-world retrieval, they must handle diverse domains, making generalization to unseen data critical. We introduce ELViS, an image-to-image similarity model that generalizes effectively to unseen domains. Unlike conventional approaches, our model operates in similarity space rather than representation space, promoting cross-domain transfer. It leverages local descriptor correspondences, refines their similarities through an optimal transport step with data-dependent gains that suppress uninformative descriptors, and aggregates strong correspondences via a voting process into an image-level similarity. This design injects strong inductive biases, yielding a simple, efficient, and interpretable model. To assess generalization, we compile a benchmark of eight datasets spanning landmarks, artworks, products, and multi-domain collections, and evaluate ELViS as a re-ranking method. Our experiments show that ELViS outperforms competing methods by a large margin in out-of-domain scenarios and on average, while requiring only a fraction of their computational cost. Code available at: https://github.com/pavelsuma/ELViS/

## 1 Introduction

Instance-level image retrieval aims to identify images of a specific object, whether a landmark, toy, painting, or product, within a large image database. The best-performing approaches rely on local descriptors (Cao et al., 2020; Tan et al., 2021; Lee et al., 2022; Zhu et al., 2023; Suma et al., 2024; Xiao et al., 2025), incorporating an image-to-image similarity model to refine a shortlist of the most similar images. This shortlist is initially retrieved using global image descriptors, often derived from foundation models (Oquab et al., 2024; Zhai et al., 2023; Radford et al., 2021).

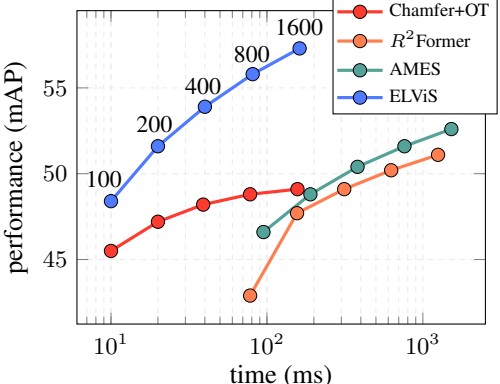

Figure 1: **Performance vs. time.** Average performance across 8 datasets and multiple domains for fixed numbers of re-ranked images indicated with text labels. All models are trained on the *landmarks* domain (GLDv2). Runtime is estimated from model latencies reported in Table 5.

Generalization to unseen domains is essential for two reasons: (i) it is inherent to retrieval, since training and test instances are disjoint, and (ii) collecting large instance-level training sets across diverse domains is notoriously challenging. Nevertheless, most methods remain confined to single-domain evaluation. Models trained on landmarks or product pairs are typically tested on benchmarks from the same domain, leaving it unclear to what extent they overfit and limiting their applicability in broader, real-world scenarios.

In this work, we challenge this paradigm by studying retrieval from a single-source domain generalization perspective, a setting mostly explored in classification (Csurka et al., 2022). We argue that in the era of foundation models, trained across diverse domains, using them off-the-shelf for both global and local descriptors is a promising strategy for cross-domain performance. Building on this, we focus on learning image-to-image similarity models for retrieval *re-ranking* that operate on sets

of local descriptors extracted from foundation models. This direction is supported by recent findings that a learnable similarity model (Suma et al., 2024) trained on frozen DINOv2 (Oquab et al., 2024) features generalizes well across domains (Kordopatis-Zilos et al., 2025), despite being trained only on landmarks and not specifically designed for generalization.

We propose an **Efficient Local Visual Similarity** model, called **ELViS**, which operates on patterns of local descriptor similarity, *i.e.* correspondence patterns, rather than on descriptors of visual appearance. This enables a more general and transferable image-level similarity measure, which is similar to observations in classical computer vision work (Shechtman & Irani, 2007). The local descriptor similarity matrix is refined using optimal transport (OT) with data-dependent gains that discard uninformative descriptors, followed by a learnable counting step that emphasizes strong correspondences. Notably, ELViS is conceptually simpler than existing methods, free of black-box modules, and built from a sequence of intuitive and interpretable steps. The architecture carries a strong inductive bias; each design choice introduces explicit priors on how to infer global similarity from local similarities. It is substantially more efficient and generalizes significantly better (see Figure 1) than prior approaches (Tan et al., 2021; Shao et al., 2023; Suma et al., 2024), which rely on heavy transformer architectures lacking priors and interpretability.

To evaluate instance-level image retrieval under domain generalization, we introduce a benchmarking protocol that unifies eight existing datasets across diverse domains: *landmarks* ($\mathcal{R}$OP+1M, GLDv2), *household items* (SOP), *retail products* (Product1M, RP2K), *artworks* (MET), and *multi-domain sets* (ILIAS, INSTRE). Benchmarks are grouped into in-domain and out-of-domain test sets depending on the training domain. To our knowledge, this is the first work to conduct such an extensive evaluation of single-source domain generalization in instance-level retrieval. Our evaluation confirms that similarity-based models generalize better than descriptor-based ones, which tend to overfit the training domain and excel only on seen distributions. With its learnable voting process and explicit mechanisms against overfitting, ELViS achieves even stronger generalization across unseen domains.

In summary, we introduce ELViS, a novel *similarity-based* re-ranking model that a) operates directly on sets of local-descriptor similarities via a novel OT formulation, b) is composed of simple, lightweight components, and c) provides a high degree of interpretability at multiple stages of the pipeline. We evaluate ELViS on eight diverse instance-level benchmarks and show that, in addition to being substantially faster, it delivers large performance gains on out-of-domain datasets while matching the performance of much heavier models on the training domain.

## 2 RELATED WORK

**Image retrieval re-ranking.** Among re-ranking methods, one line of research focuses on query expansion (Arandjelović & Zisserman, 2012; Radenović et al., 2019; Shao et al., 2023; Gordo et al., 2020), primarily using global descriptors. Another approach, which is also the focus of this work, leverages local descriptors for improved re-ranking. In the Bag-of-Words framework (Csurka et al., 2004) with hand-crafted descriptors (Lowe, 2004), a common strategy is to impose simple geometric constraints (Sivic & Zisserman, 2003) or perform RANSAC-like verification (Philbin et al., 2007). Since then, these methods have been adapted to work with local descriptors derived from deep networks (Noh et al., 2017; Simeoni et al., 2019), ultimately surpassing their hand-crafted predecessors.

Deep learning models have emerged as powerful alternatives to estimate image similarity based on local descriptor sets. State-of-the-art methods such as RRT (Tan et al., 2021) and AMES (Suma et al., 2024) match local descriptors using standard transformer-based architectures. Unlike these models, which take descriptor vectors as input, an alternative approach is to compute local descriptor similarities first, forming a *similarity-based representation* of the image pair. Early similarity-based models are employed for video retrieval, computing Chamfer similarity at both the frame and video levels and employing a 2D convolutional network to capture temporal relationships, and generalizing across various video retrieval tasks (Kordopatis-Zilos et al., 2019; 2023). For image retrieval, CVNet (Lee et al., 2022) densely computes similarities across all local descriptors and processes them with a computationally expensive 4D convolutional network. In contrast, $R^2$Former (Zhu et al., 2023) builds a similarity representation from sparse sets of local descriptors and uses a transformer architecture for similarity estimation. ELViS is also a *similarity-based* model, but it is significantly simpler, faster, and more intuitive, while promoting better generalization to unseen domains.

In the context of fine-grained sketch retrieval, Chowdhury et al. (2022) employ optimal transport to aggregate local region descriptors. In contrast to their formulation, which incorporates Lagrange multipliers for cross-modal matching, we adopt entropy-regularized OT and devise a fully differentiable counting mechanism, particularly effective for cross-domain generalization.

**Domain generalization.** Single-source domain generalization is predominantly investigated in image classification (Khosla et al., 2012; Li et al., 2017; Csurka et al., 2022). The prevailing strategies involve synthetic data generation techniques that operate by augmenting training samples in the image space (Volpi et al., 2018; Xu et al., 2021b;a) or representation space (Mancini et al., 2020; Zhou et al., 2021), or by directly generating novel samples (Yue et al., 2019; Qiao et al., 2020). Local descriptors paired with BoW also show benefits for generalization in classification (Wan et al., 2022). The generalization ability is obtained during a training process that either starts from scratch or consists of fine-tuning a network pretrained on ImageNet.

In tasks such as image matching (Jin et al., 2020) and 3D reconstruction (Schönberger & Frahm, 2016), where open-world performance and generalization are essential, we observe a distinct trend compared to other computer vision tasks. Hand-crafted representations (Lowe, 2004) and matching methods (Schönberger & Frahm, 2016) remain among the top-performing approaches. A major shift happens with the advent of large pre-trained foundation models (An et al., 2023; Zhai et al., 2023; Oquab et al., 2024; Radford et al., 2021). These models are exposed to vast amounts of data during training, making it unclear whether a given test image truly belongs to an unseen domain. Notably, keeping their representations frozen while applying hand-designed methods has proven highly effective across diverse object types and domains (Örnek et al., 2024). While training a model on top of frozen representations may introduce domain dependence, carefully designed approaches have been shown to encourage generalization (Jiang et al., 2024).

## 3 METHOD

In this section, we introduce **ELViS**, an image-to-image similarity method that takes sets of local descriptors as input. Instead of operating directly on the descriptors, our approach builds, refines, and processes their similarity matrix, and enables a learnable and intuitive voting mechanism with few parameters that generalizes well to unseen domains. An overview is presented in 2.

### 3.1 BACKGROUND

**Problem formulation.** The goal of an image retrieval system is to search a database $\mathcal{D}$ using a query image $q$ and retrieve the most relevant images. At its core, image retrieval depends on a pairwise image-to-image similarity function $s(q, x) \in \mathbb{R}$, which measures the relevance between the query $q$ and each database image $x \in \mathcal{D}$, allowing for ranking based on similarity scores. We aim to learn $s$ by training on a source domain, typically rich in instance-level training data, and then test on a target domain that remains unseen during training.

**Local descriptors.** After an initial ranking with global descriptors, state-of-the-art instance-level retrieval methods include a second-stage pairwise re-ranking step using local descriptors (Suma et al., 2024; Tan et al., 2021; Zhu et al., 2023). In ViT architectures (Dosovitskiy et al., 2021), these local descriptors correspond to a subset of the patch descriptors. Given an image $x$, the set of local descriptors is represented as a $D' \times M$ matrix, $\mathbf{X} = [\mathbf{x}_1 \dots \mathbf{x}_i \dots \mathbf{x}_M]$, where each descriptor $\mathbf{x}_i \in \mathbb{R}^{D'}$ is a $D'$-dimensional vector. We select the strongest $M$ descriptors per image based on a strength score (Suma et al., 2024). For efficiency and better task adaptation, the descriptor dimensionality is reduced from $D'$ to $D$ through a *learnable linear projection*. This projection is implemented as a linear layer followed by layer normalization and $\ell_2$-normalization per local descriptor. The projection is a common component among all learnable methods we compare with in the experiments.

**Image similarity.** The similarity $s(q, x) \in \mathbb{R}$ between images $q$ and $x$ is computed as a function of their corresponding local descriptor matrices $\mathbf{Q}, \mathbf{X} \in \mathbb{R}^{D \times M}$, *i.e.*, $s(q, x) := s(\mathbf{Q}, \mathbf{X})$. The core of $s$ processes the *local descriptor similarity matrix* $\mathbf{S} = \mathbf{Q}^\top \mathbf{X} \in \mathbb{R}^{M \times M}$. For notational clarity, we assume the same number of descriptors per image for $q$ and $x$, while the method is generic and does not impose such a constraint.

Figure 2: **Detailed overview of ELViS.** The similarity matrix is refined using optimal transport with descriptor-dependent dustbin gains. The strongest local *similarities* per descriptor are then selected and transformed element-wise by a learned function $f$, before being sum-aggregated into a scalar global similarity. During training, a modified BCE loss with a learnable function $g$ reshapes the penalty curve; $g$ is used only for training and is expandable at inference.

## 3.2 FROM LOCAL DESCRIPTORS TO LOCAL SIMILARITIES

The proposed approach operates in the space of *descriptor similarities*, in particular similarity matrix **S**, whose values represent correspondences between the patches the descriptors are extracted from. We introduce a refinement of **S** to generate matrix **S'** that emphasizes mutually consistent strong correspondences and discards correspondences from uninformative descriptors.

We formulate the problem as a variant of optimal transport which is efficiently solved using the iterative Sinkhorn-Knopp algorithm (Sinkhorn & Knopp, 1967) allowing back-propagation through the optimization process. More precisely, our objective is to find a matrix **P**, that maximizes $\langle \mathbf{P}, \mathbf{S} \rangle_F$ subject to constraints $\mathbf{P1}_M = \mathbf{1}_M$ and $\mathbf{P}^\top \mathbf{1}_M = \mathbf{1}_M$, where $\mathbf{1}_M$ is a vector of ones of size $M$, and $\langle \cdot, \cdot \rangle_F$ denotes the Frobenius inner product. [1] The solution **P** is seen as a refined, doubly stochastic, similarity matrix. To allow distracting or uninformative descriptors (*e.g.* those extracted from the background) to be ignored and excluded from the final correspondence matrix, we introduce slack variables that indicate the gain of not transporting mass for a given descriptor. This is what SuperGlue[2] refers to as *dustbins* (Sarlin et al., 2020), *i.e.* the gain of assigning a descriptor to the dustbin and not to any descriptor in the other image. It is achieved by creating an augmented $(M + 1) \times (M + 1)$ matrix $\hat{\mathbf{S}}$ by

$$\hat{\mathbf{S}} = \begin{bmatrix} \mathbf{S} & \mathbf{u} \\ \mathbf{v}^\top & \omega \end{bmatrix}, \tag{1}$$

where $\mathbf{u}, \mathbf{v} \in \mathbb{R}^M$ contain the dustbin gains for the query and database image descriptors, respectively, while $\omega$ accounts for the gain related to the total mass moved to the dustbins.

We define **P** as the solution to the following optimization problem:

$$\max_{\mathbf{P}} \langle \mathbf{P}, \hat{\mathbf{S}} \rangle_F + \lambda H(\mathbf{P}) \tag{2}$$

$$\text{s.t.} \quad \mathbf{P1}_{M+1} = \mathbf{a}, \qquad \mathbf{P}^\top \mathbf{1}_{M+1} = \mathbf{b},$$

where $\mathbf{a} = [\mathbf{1}_M^\top \quad M]^\top$ and $\mathbf{b} = [\mathbf{1}_M^\top \quad M]^\top$ are the marginal constraints extended to include dustbins. We use the entropy-regularized variant of Sinkhorn-Knopp (Cuturi, 2013), with regularization term $\lambda$. After optimization, we drop the additional dustbin row and column and maintain the *refined similarity matrix* $\mathbf{S}' = \mathbf{P}_{1:M,1:M}$ for the following steps.

**Descriptor-dependent dustbin gains.** Prior work (Sarlin et al., 2020) sets dustbin gains **u**, **v** to a fixed or learnable scalar. Instead, we predict the gain based on the descriptor itself with function $h : \mathbb{R}^D \to \mathbb{R}$. The gains are given by

$$\begin{aligned} \mathbf{u} &= [u_1 \dots u_i \dots u_M] & &= [h(\mathbf{q}_1) \dots h(\mathbf{q}_i) \dots h(\mathbf{q}_M)] \\ \mathbf{v} &= [v_1 \dots v_i \dots v_M] & &= [h(\mathbf{x}_1) \dots h(\mathbf{x}_i) \dots h(\mathbf{x}_M)], \end{aligned} \tag{3}$$

---

[1] Note that we operate with a similarity matrix and not a cost matrix, therefore the maximization instead of minimization. Similarity is seen as negative cost, or as the gain of transporting mass.

[2] Prior work applies Sinkhorn-Knopp on similarity matrices to establish point correspondences, while we care about the correspondence strengths and aim to aggregate them into an image-level similarity score.

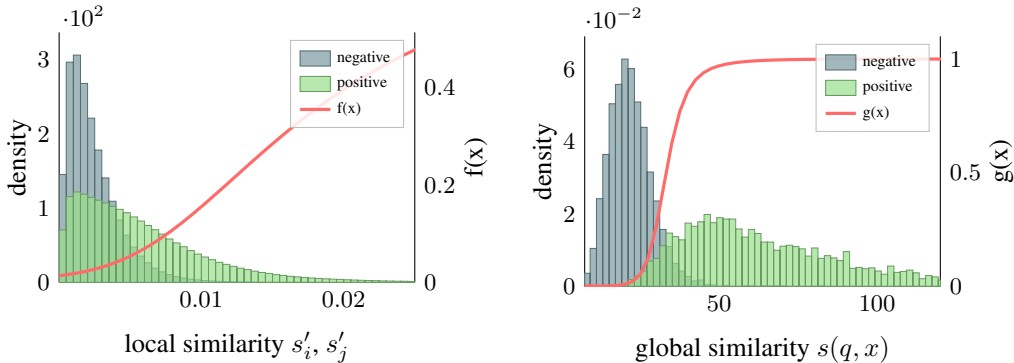

Figure 3: **Shape of the learned univariate functions $f$ (left) and $g$ (right).** Although parameterized as MLPs, both functions learn well-behaved scalar transformations that effectively separate matching and non-matching distributions. The distributions of input values are visualized separately for positive and negative image pairs, sampled during training.

where $u_i$ and $v_i$ denote the $i$-th element of vector $\mathbf{u}$ and $\mathbf{v}$, respectively. Larger dustbin gains of $u_i$ and $v_i$ assign higher chance for the correspondences of descriptor $i$ to be moved to the dustbin. We implement $h$ as a two-layer MLP with a GELU activation function (Hendrycks & Gimpel, 2016). Gain $\omega$ is a learnable scalar. Our experiments demonstrate that using descriptor-dependent dustbin gains is essential for the effectiveness of such a refinement step in the overall pipeline.

### 3.3 FROM LOCAL SIMILARITIES TO GLOBAL SIMILARITY

In this step, we transform the similarity matrix $\mathbf{S}'$ into a set of votes that are aggregated into a single value representing the global similarity of the input image pair.

**Strongest vote per descriptor.** Given matrix $\mathbf{S}'$, which contains similarities for all pairs of descriptors, we keep the strongest similarity for each descriptor of each of the two images, acting as a *vote*. This is equivalent to selecting the strongest correspondence per descriptor. Formally, this is given by

$$s_i' = \max_{j \in \{1,\ldots,M\}} \mathbf{S}'_{i,j}, \qquad s_j' = \max_{i \in \{1,\ldots,M\}} \mathbf{S}'_{i,j}, \qquad \forall i, j \in \{1, \ldots, M\}, \qquad (4)$$

where $s_i'$ and $s_j'$ are row- and column-wise max-pooled similarities[3]. Summing all similarities in $s_i'$ and $s_j'$ jointly, for $i, j = 1 \ldots M$, is equivalent to computing Chamfer similarity on $\mathbf{S}'$ under the assumption of equal descriptor set cardinalities. We go one step further in the next processing stage. It is worth noting that Chamfer similarity after vanilla optimal transport, even without learning, already serves as a strong baseline for generalization, as confirmed by our experiments, which motivates our choice to build on and extend this architecture.

**Learnable vote strength and counting.** We transform votes $s_i'$ and $s_j'$ via function $f : \mathbb{R} \to \mathbb{R}$, a real function mapping an input scalar similarity to an updated scalar similarity, *i.e.* vote, in $[0, 1]$. Function $f$ is implemented by a two-layer MLP with GELU activations and sigmoid at its output. The image-to-image similarity is then computed by counting all votes via summation

$$s(q, x) = \sum_{i=1}^{M} f(s_i') + \sum_{j=1}^{M} f(s_j'). \qquad (5)$$

This voting-based global similarity estimation is inspired by classical works in image retrieval (Tolias & Jégou, 2014), which demonstrate that the number of strong local descriptor correspondences

---

[3]We equivalently define $s_i$ and $s_j$ from max pooling in $\mathbf{S}$, which is only used for visualization purposes.

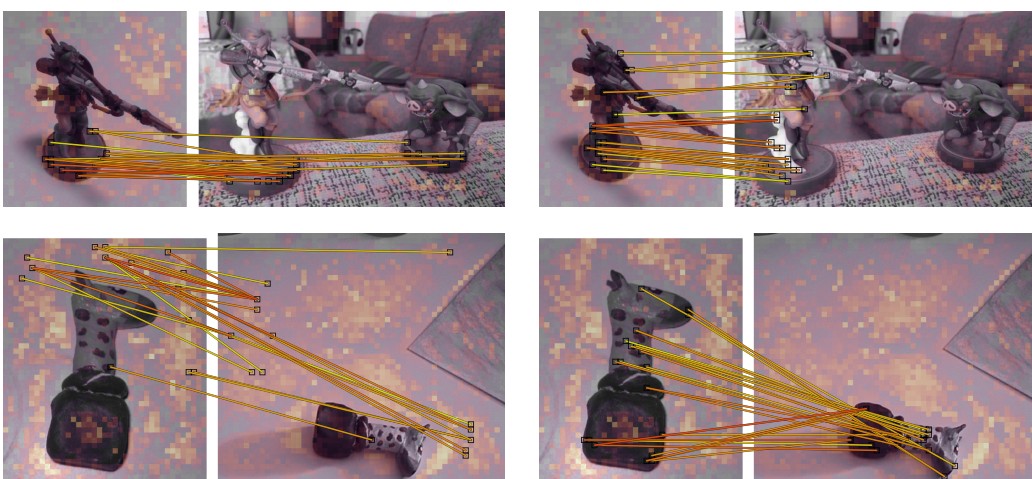

Figure 4: **Visualization of the 25 strongest correspondences (votes)** among $s_i, s_j$ (left) and $s_i', s_j'$ (right) before (left) and after (right) refinement with optimal transport. Red (yellow) represents high (low) similarity. Raw similarity values in $\mathbf{S}$ (left) and values in $\mathbf{S}'$ after passing them through $f$ (right) are used. Heatmaps represent the dustbins values by evaluating $h$ densely for all patches in both images; bright values indicate large dustbin gain and uninformative descriptors.

is a robust indicator of image similarity. In contrast to hand-crafted weighting functions for correspondence strengths, like RBF-kernel (Jégou et al., 2008) or monomial kernel (Tolias et al., 2013), our learnable function $f$ adaptively transforms the similarities to optimize retrieval performance. We visualize the learned $f$ after training in Figure 3, which noticeably differentiates from linear weighting, *i.e.* identity function is $f$ and the corresponding MLP would not be included in the model. Our experiments show that using a learnable $f$ is beneficial for generalization; excluding it makes the descriptor projection layer responsible for obtaining appropriate correspondence strength and the method more descriptor-dependent and domain-dependent.

**Example visualization.** Figure 4 shows the strongest correspondences selected from the similarity matrices before and after refinement, *i.e.* from $\mathbf{S}$ and $\mathbf{S}'$, respectively. Without refinement, many strong correspondences are formed between background or non distinctive regions; the refinement step suppresses these mainly due to the use of dustbin gains. The final set contains a large number of correct object correspondences, whose strengths are meaningfully transformed by $f$. Summing these strengths yields the final similarity score between the two images, making the process both intuitive and interpretable.

### 3.4    TRAINING AND INFERENCE

**Training.** We introduce a data-adapted variant of the Binary Cross-Entropy loss (BCE) to train ELViS with positive and negative image pairs. Standard BCE minimizes $-\log p$ for positive pairs and $-\log(1-p)$ for negative pairs, where $p = s(q, x)$ is the predicted similarity. In our formulation, $p$ is first passed through a learnable function $g : \mathbb{R} \rightarrow [0, 1]$ before BCE is applied, yielding losses $-\log g(p)$ and $-\log(1 - g(p))$. This modification no longer optimizes the log-likelihood of the predicted probability but instead the log of a transformed version of it. By reshaping the penalty curve through $g$, we control which prediction errors are emphasized or downweighted. The function $g$ is implemented as a two-layer MLP with a sigmoid output, and its learned shape is shown in Figure 3. Empirically, $g$ tends to be nearly piecewise-linear, with its slope changing around the region where positive and negative pairs start to overlap more, thereby enabling differentiated penalization of errors. Thus, training proceeds under a warped notion of similarity defined by $g$.

**Inference and re-ranking with ELViS.** At inference, the auxiliary function $g$ is discarded. This strategy parallels the use of projection heads in self-supervised learning (Chen et al., 2020; Zbontar et al., 2021) that are expendable modules used for optimization, encouraging generalization to other tasks. Discarding $g$ is valid because, similar to a learnable temperature in contrastive losses (Radford et al., 2021), $g$ scales the similarity for the loss without altering the ranking order, provided it is an

increasing function. Although monotonicity is not enforced during training, we consistently observe $g$ to be monotonic, where it matters, in practice. Attempts to enforce monotonicity explicitly, *e.g.* by constraining MLP weights to be non-negative as in (You et al., 2017), slightly degrade performance and require further exploration.

Given a ranked list of candidate images for a query, obtained for instance using a global-representation-based retrieval method, we form query–candidate pairs and apply ELViS to compute refined similarity scores, which are then used to re-rank the list.

## 4 EXPERIMENTS

### 4.1 EXPERIMENTAL SETUP

**Datasets.** We evaluate the proposed method and the most related approaches on 8 datasets containing instance-level annotations and spanning multiple domains: (i) *Landmarks* – $\mathcal{R}$Oxford and $\mathcal{R}$Paris, reported jointly as $\mathcal{R}$OP+1M (Philbin et al., 2007; 2008; Radenović et al., 2018), and GLDv2 (Weyand et al., 2020); (ii) *Household items* – SOP (Song et al., 2016); (iii) *Retail products* – Product1M (Zhan et al., 2021) and RP2K (Peng et al., 2020); (iv) *Artworks* – MET (Ypsilantis et al., 2021); and (v) *Multi-domain* – ILIAS (Kordopatis-Zilos et al., 2025) and INSTRE (Wang & Jiang, 2015). Further details are found in the Appendix A.

**Training domains.** We go beyond the standard evaluation commonly adopted in instance-level retrieval papers, *i.e.* training on GLDv2 due to its large number of images and instances and evaluating on the same domain, *e.g.* GLDv2 and $\mathcal{R}$OP+1M. We put a focus on generalization and introduce a protocol consisting of 8 instance-level retrieval datasets spanning diverse visual domains. Depending on which dataset is used for training the re-ranking models, we categorize the datasets into two groups: in-domain and out-of-domain, and report the average performance separately for each. We select two large datasets with clearly defined train/test splits as training domains: GLDv2 and SOP. When training on GLDv2, we consider its test set, and $\mathcal{R}$OP+1M, as in-domain testing and the remaining 6 datasets as out-of-domain. When training on SOP, we consider only its test set as in-domain testing, while the remaining 7 datasets are treated as out-of-domain.

**Evaluation protocol.** Retrieval performance is measured using mean Average Precision (mAP) on $\mathcal{R}$OP+1M and mAP@K on the rest of the datasets. We additionally report average performance over all datasets in each group, *i.e.*, in-domain and out-of-domain. We re-rank the top $400$ retrieved images in all experiments. As in AMES, we select the $M = 600$ strongest descriptors according to a local feature detector (Noh et al., 2017; Tolias et al., 2020) trained with images from the corresponding domain. We extract local descriptors with DINOv2 (Oquab et al., 2024), as in AMES, as well as DINOv3 (Siméoni et al., 2025) and SigLIP2 (Tschannen et al., 2025).

**Compared methods.** We compare the performance of the proposed ELViS with the most relevant re-ranking approaches from the literature, namely Reranking Transformer (RRT) (Tan et al., 2021), $R^2$Former (Zhu et al., 2023), and AMES (Suma et al., 2024). We also compare to hand-crafted Chamfer similarity (Barrow et al., 1977; Razavian et al., 2016) applied directly on $\mathbf{S}$, and on the refined matrix $\mathbf{S}'$ obtained via vanilla optimal transport (OT) with fixed dustbin gains equal to 1. Both these methods serve as baselines for local descriptor performance without training a re-ranking model. ELViS internally performs a similar process to matching correspondences, thus we also compare to established feature matching models in the Supplementary material (Section D). Retrieval using only global representation is also evaluated, *i.e.*, there is no use of re-ranking. We train and evaluate all methods using the publicly available AMES repository[4], and integrate the official implementations provided by the authors of RRT[5], and $R^2$Former[6].

---

[4]https://github.com/pavelsuma/ames
[5]https://github.com/uvavision/RerankingTransformer
[6]https://github.com/Jeff-Zilence/R2Former

| Method | $\mathcal{R}$OP+1M | GLDv2 | ILIAS | INSTRE | MET | Prod1M | RP2K | SOP-1k | ID | OOD | avg |
|---|---|---|---|---|---|---|---|---|---|---|---|
| | | | | landmarks domain (GLDv2) | | | | | | | |
| **No re-ranking** | 57.7 | 27.3 | 9.4 | 65.3 | 61.6 | 24.7 | 39.0 | 33.7 | 42.5 | 38.9 | 39.8 |
| **Chamfer**[†] | 56.7 | 23.8 | 6.2 | 55.0 | 37.3 | 17.8 | 55.8 | 48.6 | 40.2 | 36.8 | 37.6 |
| **Chamfer+OT**[†] | 60.6 | 23.8 | 14.3 | 76.0 | 74.0 | 35.1 | 55.4 | 46.2 | 42.2 | 50.2 | 48.2 |
| **RRT**[⋆] | 69.2 | 33.1 | 13.1 | 72.4 | 64.1 | 29.3 | **60.7** | 52.1 | 51.1 | 48.6 | 49.2 |
| $R^2$**Former**[†] | 68.5 | 32.6 | 15.2 | 77.6 | 72.0 | 35.6 | 47.7 | 43.7 | 50.6 | 48.6 | 49.1 |
| **AMES**[⋆] | **70.1** | **34.7** | 14.6 | 75.6 | 70.7 | 32.3 | 56.5 | 48.5 | **52.4** | 49.7 | 50.4 |
| **ELViS**[†] | 68.8 | 32.2 | **18.8** | **80.4** | **77.9** | **41.5** | 59.2 | 52.3 | 50.5 (-1.9) | **55.0** (+4.8) | **53.9** (+3.2) |
| | | | | household items domain (SOP) | | | | | | | |
| **No re-ranking** | 57.7 | 27.3 | 9.4 | 65.3 | 61.6 | 24.7 | 39.0 | 33.7 | 33.7 | 40.7 | 39.8 |
| **Chamfer**[†] | 46.2 | 15.4 | 6.7 | 63.2 | 45.2 | 24.7 | 50.3 | 50.2 | 50.2 | 35.9 | 37.7 |
| **Chamfer+OT**[†] | 52.3 | 18.6 | 11.7 | 75.9 | 71.6 | 37.5 | 52.7 | 45.8 | 45.8 | 45.7 | 45.7 |
| **RRT**[⋆] | 43.4 | 10.8 | 12.2 | 68.4 | 25.5 | 32.2 | 46.9 | 57.1 | 57.1 | 34.2 | 37.1 |
| $R^2$**Former**[†] | 55.5 | **23.7** | 12.9 | 73.4 | 59.3 | 32.8 | 42.0 | 51.1 | 51.1 | 42.8 | 43.8 |
| **AMES**[⋆] | 55.6 | 17.2 | 12.4 | 72.2 | 44.2 | 36.7 | 51.8 | 56.7 | 56.7 | 41.4 | 43.3 |
| **ELViS**[†] | **59.7** | 22.9 | **18.6** | **81.1** | **76.7** | **44.1** | **54.2** | 54.9 | 54.9 (-2.2) | **51.0** (+5.3) | **51.5** (+5.8) |

Table 1: **Domain generalization performance (mAP)**. Training performed either on *landmarks* (GLDv2) or *household items* (SOP). Results reported per dataset and as average over in-domain (ID), out-of-domain (OOD), and all datasets (avg). Local descriptors extracted with DINOv2 (Oquab et al., 2024). Gray indicates in-domain results. Green (Red) highlights gain (loss) of ELViS over the second best method. †, ⋆ indicate similarity-based and descriptor-based models, respectively.

| Method | ID | OOD | avg |
|---|---|---|---|
| **No re-ranking** | 49.6 | 60.5 | 57.8 |
| **Chamfer** | 42.3 | 44.5 | 44.0 |
| **Chamfer+OT** | 41.7 | 51.4 | 49.0 |
| **RRT** | 57.1 | 56.2 | 56.4 |
| $R^2$**Former** | 56.0 | 63.9 | 61.9 |
| **AMES** | **58.6** | 62.8 | 61.8 |
| **ELViS** | 56.9 (-1.7) | **67.4** (+3.5) | **64.8** (+2.9) |

(a) DINOv3 (Siméoni et al., 2025)

| Method | ID | OOD | avg |
|---|---|---|---|
| **No re-ranking** | 25.4 | 57.8 | 49.7 |
| **Chamfer** | 32.8 | 58.7 | 52.2 |
| **Chamfer+OT** | 29.3 | 62.3 | 54.1 |
| **RRT** | **37.5** | 58.4 | 53.2 |
| $R^2$**Former** | 35.2 | 63.0 | 56.0 |
| **AMES** | 37.1 | 62.7 | 56.3 |
| **ELViS** | 36.4 (-1.1) | **68.7** (+5.7) | **60.6** (+4.3) |

(b) SigLIP2 (Tschannen et al., 2025)

Table 2: **Performance (mAP) comparison using local descriptors from additional foundational models**. Training performed on *landmarks* (GLDv2). Results reported per dataset and as averages over in-domain (ID), out-of-domain (OOD), and all datasets (avg). Green (Red) highlights gain (loss) of ELViS over the second best method.

## 4.2 RESULTS

**Performance comparison.** We present a performance comparison using DINOv2 and two different training sets in Table 1 and DINOv3 and SigLIP2 and training on landmarks in Table 2. We maintain backbone consistency between local and global similarity, *i.e.* the same model is used for initial global retrieval and re-ranking with local descriptors. We identify the following key observations:

(i) *ELViS achieves the best average performance overall.* Across all settings, ELViS outperforms all other methods in terms of mean average precision by a significant margin, ranging from 2.9 to 5.8, compared to the second best approach.

(ii) *ELViS excels at domain generalization.* The performance gains for OOD datasets are large, *i.e.* improvements over the second best method equal to 4.8 and 5.3 while training on landmarks and household items, respectively, using DINOv2, and 3.5 and 5.7 using DINOv3 and SigLIP2, respectively, while training on landmarks.

(iii) *ELViS provides significant gains on harder datasets.* This is particularly evident in the case of the recent ILIAS datasets, which feature a database of *over 100M images*. Here, the *relative* performance improvement of ELViS over the second-best method is over 23% and 36% while training on landmarks and household items, respectively.

| Method | | ID | OOD | avg |
|---|---|---|---|---|
| No re-ranking | | 42.5 | 38.9 | 39.8 |
| ELViS | | **50.5** | **55.0** | **53.9** |
| w/o | dustbin | 23.1 (-27.4) | 32.8 (-22.2) | 30.4 (-23.5) |
| w/o | descriptor-dependent gain | 48.8 (-1.7) | 52.4 (-2.6) | 51.5 (-2.4) |
| w/o | $f$ function | 50.8 (+0.3) | 53.8 (-1.2) | 53.1 (-0.8) |
| w/o | $g$ function | 45.6 (-4.9) | 49.5 (-5.5) | 48.5 (-5.4) |
| w/o | $f, g$ functions | 47.3 (-3.2) | 48.5 (-6.5) | 48.2 (-5.7) |
| w/o | descriptor projection | 48.4 (-2.1) | 51.7 (-3.3) | 50.8 (-3.1) |

Table 3: **Ablation study on method components**. Average, ID, and OOD performance, when training on landmarks. Green (Red) highlights gain (loss) over ELViS.

(iv) *Similarity-based models are more robust in OOD but weaker in ID*. This trend extends beyond ELViS; all similarity-based models seem to be top performing in OOD, but lag slightly behind in ID. For example, ELViS performs about 1-2 mAP worse than AMES on ID. Transformer models operating on local descriptors effectively overfit to the training domain, while similarity-based models generalize due to strong inductive biases.

(v) *Hand-crafted similarity on top of strong foundational model representations is a strong baseline for OOD*. Chamfer+OT is the second best performing method across 3 out of 4 settings in OOD. This supports our choice of extending such architecture with minimal learnable parts that significantly boost performance without compromising speed (Figure 1) or interpretability. Note that Chamfer by itself is not an effective re-ranking strategy, indicating the value of OT and similarity refinement.

**Ablations.** In Table 3, we present an ablation study of the proposed approach, analyzing the contribution of its internal components. Naively applying OT without dustbins (trained/tested only for an equal number of descriptors for both images) leads to a severe performance drop because uninformative descriptors are not ignored. Learning a scalar gain for all descriptors, as in Sarlin et al. (2020), degrades performance and demonstrates the value of our contribution. Interestingly, when removing function $f$, with or without the presence of $g$, the model relies directly on input descriptors to form vote strengths, therefore encouraging overfitting to the training domain, which improves ID performance at the expense of OOD generalization. Function $g$ is essential for effective training of ELViS, and its introduction results in a noticeable boost. Lastly, we additionally evaluate the impact of the descriptor projection as the earlier learnable layer, which gives a boost on both ID and OOD.

**Complexity analysis.** Table 5 presents the computational complexity of ELViS compared to the best competitors. ELViS is the most lightweight and fastest model, containing the fewest network parameters, *i.e.* about $2\times$ fewer than $R^2$Former and about $20\times$ fewer than AMES and RRT. Importantly, ELViS is several times faster. As further illustrated in Figure 1, this efficiency enables ELViS to re-rank significantly more images, yielding an additional performance boost over other methods if we consider a fixed time budget. Notably, compared to the Chamfer+OT baseline, ELViS is as fast, while its newly added learnable components, *i.e.* data-dependent dustbin gains, functions $f$ and $g$, and the descriptor projection, give a strong performance boost. Note that when measuring latency, the projected local descriptors and dustbin gains for database images are considered precomputed and stored. Obtaining these components for the query image is a constant cost that does not depend on the number of images to re-rank, and is therefore excluded from the reported times.

**Improved in-domain performance with hybrid architecture.** ELViS excels in unseen domains, yet it is weaker than the SotA descriptor-based approaches in the seen domain. To reduce this gap, we design a *hybrid* model combining AMES and ELViS. We replace the standard descriptor projection used in ELViS with a sequence of AMES transformer blocks. The two input descriptor sets are passed through this AMES module, comprising several self- and cross-attention layers. The resulting transformer outputs are treated as refined descriptor sets for each image, which are then passed to ELViS. We train this hybrid model end-to-end using the default ELViS parameters. Table 4 shows the combination significantly boosts ID performance, with only a slight compromise in OOD.

| Method | ID | OOD | avg |
|---|---|---|---|
| No re-ranking | 42.5 | 38.9 | 39.8 |
| AMES | **52.4** | 49.7 | 50.4 |
| ELViS | 50.5 | **55.0** | 53.9 |
| ELViS + AMES | 52.1 | 54.7 | **54.0** |

| Method | Params (K) | Latency ($\mu$s) |
|---|---|---|
| Chamfer+OT | 0 | 98 |
| RRT | 2232 | 656 |
| $R^2$Former | 202 | 782 |
| AMES | 2130 | 952 |
| ELViS | **96** | **101** |

Table 4: **Hybrid architecture combining descriptor-based and similarity-based processing.** In the hybrid model, AMES performs intra-image and inter-image descriptor processing with 5 transformer blocks, then the refined output tokens are subsequently fed into ELViS.

Table 5: **Computational complexity.** Parameters include all learnable components and descriptor projection. Latency measures the average similarity estimation time per image pair (batch size 500), excluding constant costs for query and precomputed database extraction.

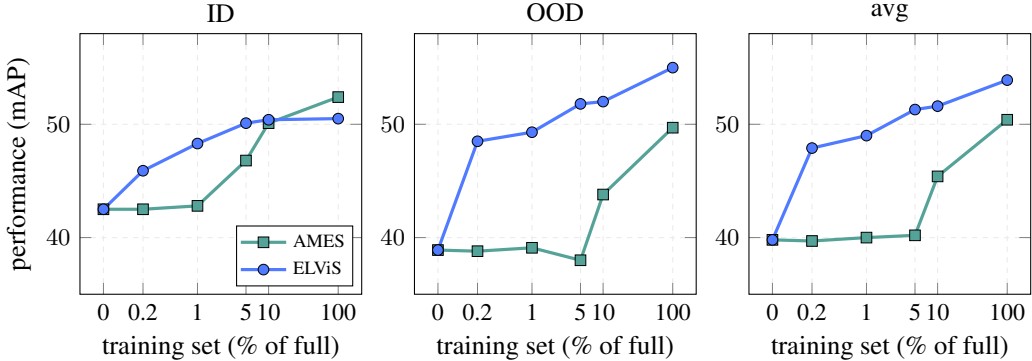

Figure 5: **Effectiveness with small training sets.** Performance of ELViS and AMES when trained on subsets of the full training set (754k images), ranging from 0.2% to 100%. No re-ranking performance is indicated at 0 training set size. Training hyperparameters are tuned for each subset size based on validation performance.

**Training with smaller training sets.** To assess the data efficiency of ELViS, we simulate limited-data scenarios by training on random subsets of our full training set from GLDv2. We evaluate models trained on 1.4k (50), 8.1k (250), 40k (1250), and 80k (2500) images (classes), corresponding to roughly 0.2%, 1%, 5%, and 10% of the default training set. As shown in Figure 5, ELViS demonstrates strong generalization even in low-data regimes. With only 1.4k images, ELViS already outperforms the global baseline with a substantial margin. Furthermore, we observe distinct scaling behaviors for ID and OOD, *i.e.* while ID performance saturates relatively early, OOD performance continues to improve. This suggests that smaller datasets are sufficient for in-domain retrieval, whereas learning robust, transferable similarity patterns requires larger-scale training. Finally, AMES starts to improve global descriptor performance only with at least 80k images, highlighting the data efficiency of ELViS.

## 5 CONCLUSION

We introduce ELViS, a lightweight and highly effective image-to-image similarity model that achieves state-of-the-art re-ranking performance across multiple instance-level retrieval benchmarks. As a similarity-based model, ELViS benefits from strong inductive biases, enabling better generalization to unseen domains compared to descriptor-based approaches. Moreover, ELViS is composed of a sequence of intuitive processing steps. By avoiding deep stacks of generic neural blocks, the model offers both improved interpretability and exceptional efficiency. In fact, it processes nearly an order of magnitude more images than the second-best model across all datasets in the same amount of time.

**Acknowledgements.** This work was supported by the Junior Star GACR (grant no. GM 21-28830M), CTU in Prague (grant no. SGS23/173/OHK3/3T/13), Horizon MSCA-PF (grant no. 101154126), and the Czech National Recovery Plan—CEDMO 2.0 NPO (MPO 60273/24/21300/21000) provided by the Ministry of Industry and Trade. We acknowledge VSB – Technical University of Ostrava, IT4Innovations National Supercomputing Center, Czech Republic, for awarding this project (OPEN-35-4) access to the LUMI supercomputer, owned by the EuroHPC Joint Undertaking, hosted by CSC (Finland) and the LUMI consortium through the Ministry of Education, Youth and Sports of the Czech Republic through the e-INFRA CZ (grant ID: 90254).

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

# Appendix

| dataset | train set | validation set | | test set | | domain | evaluation metric |
|---|---|---|---|---|---|---|---|
| | | queries | database | queries | database | | |
| GLDv2 | 754K | 379 | 762K | 750 | 762K | landmark | mAP@100 |
| SOP | 48.9K | 1K | 10.6K | 1K | 60.5K | household | mAP@100 |
| $\mathcal{R}$Oxford | – | – | – | 70 | 5K+1M | landmark | mAP |
| $\mathcal{R}$Paris | – | – | – | 70 | 6.3K+1M | landmark | mAP |
| Product1M | – | – | – | 6.2K | 38.7K | retail | mAP@100 |
| RP2K | – | – | – | 10.9K | 10.9K | retail | mAP@100 |
| MET | – | – | – | 1K | 397K | artwork | mAP@100 |
| INSTRE | – | – | – | 1.2K | 27K | multi | mAP@100 |
| ILIAS | – | – | – | 1K | 100M | multi | mAP@1K |

Table F: **Dataset statistics and metrics used**. For each dataset, we use the most commonly used metric. Train and validation set statistics are reported only for the two training datasets that are used in this work.

## A   Dataset details

We evaluate the proposed method and the closest related approaches on eight datasets containing instance-level or fine-grained recognition annotations. These datasets span multiple domains listed below. Examples of each are visualized in Figure F.

**Landmarks.** The $\mathcal{R}$Oxford (Philbin et al., 2007; Radenović et al., 2018), $\mathcal{R}$Paris (Philbin et al., 2008; Radenović et al., 2018), and Google Landmarks Dataset v2 (GLDv2) (Weyand et al., 2020) are designed for instance-level retrieval and recognition. For training, we use the same subset of the training split of GLDv2 as in AMES (Suma et al., 2024). As usual, we evaluate the *medium* and *hard* settings of the $\mathcal{R}$Oxford and $\mathcal{R}$Paris datasets together with 1M accompanying distractor images, denoted as $\mathcal{R}$OP+1M.

**Household items.** Stanford Online Products (SOP) (Song et al., 2016) is an instance-level dataset of furniture and electric appliance images sourced from eBay. It has been widely used for fine-grained image classification and contains a training/test split. For evaluation on SOP, we sample 1k test images that serve as queries. The entire test set is used for the database. The training images are further divided into a training set and a validation set in an 80%-20% split.

**Retail products.** Product1M (Zhan et al., 2021) and RP2K (Peng et al., 2020) are datasets containing a large variety of retail products, *e.g.* cosmetics and grocery store items. The former was made for instance-level retrieval, while RP2K originally targeted fine-grained image classification. We adopt its repurposed version from (Ypsilantis et al., 2023), tailored for retrieval.

**Artworks.** The MET (Ypsilantis et al., 2021) dataset depicts artworks from the Metropolitan Museum of Art in New York and is designed for instance-level recognition. To adapt the benchmark for retrieval, we keep only one positive image per query that is guaranteed to have visual overlap with it.

**Multi-domain datasets.** Instance-Level Image retrieval At Scale (ILIAS) (Kordopatis-Zilos et al., 2025) and INSTance-level visual object REtrieval and REcognition (INSTRE) (Wang & Jiang, 2015) datasets are designed for instance-level retrieval and include images from various domains, *e.g.* landmarks, products, and art.

## B   Implementation Details

We follow the standard practice of training image-to-image similarity models with local descriptors (Tan et al., 2021; Suma et al., 2024). All the learnable parameters of ELViS are trained with binary cross-entropy loss, where the ground truth label of the image pair denotes whether the two

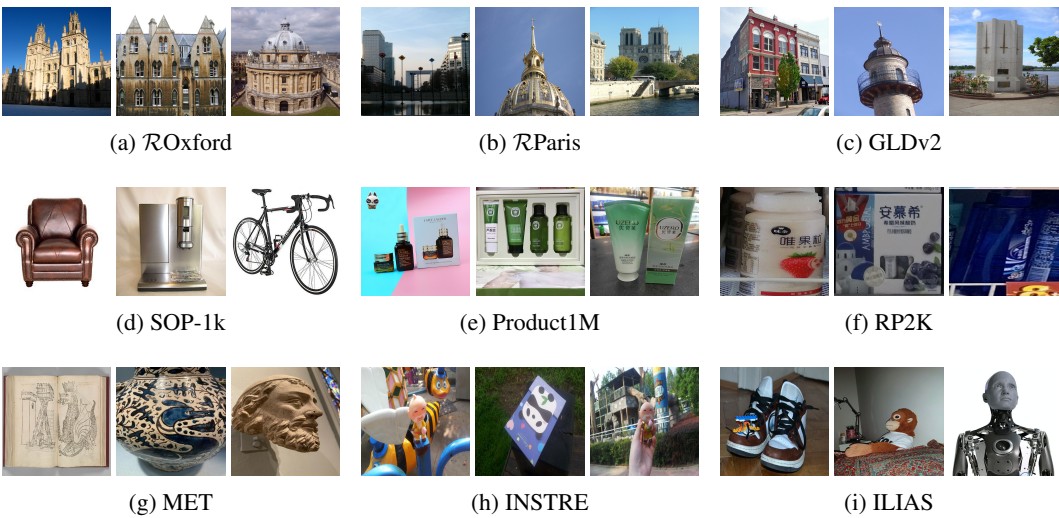

(a) $\mathcal{R}$Oxford       (b) $\mathcal{R}$Paris       (c) GLDv2

(d) SOP-1k       (e) Product1M       (f) RP2K

(g) MET       (h) INSTRE       (i) ILIAS

Figure F: **Benchmark examples**. Three random samples are shown for each of the 9 datasets. All images are resized to a square aspect ratio for better visualization.

images are positive, *i.e.* depict the same object instance, or not. We apply balanced sampling so the network sees the same number of both labels per batch. We mine challenging pairs by exploiting the most similar images as indicated by a given global descriptor representation.

Our model consists of a function $f$ implemented as an MLP with a hidden dimensionality of 16, and a function $g$, also an MLP, with a dimensionality set to 64. For the refinement step, we set $\lambda$ to 0.1 and initialize $\omega$ with 1. We run 10 iterations of the Sinkhorn-Knopp algorithm, which is a trade-off that balances the model speed and performance. For the dustbin function $h$, we retain the input dimensionality $D$ for the hidden layer. Local descriptors are extracted from the DINOv2-Base model with registers (Oquab et al., 2024; Darcet et al., 2024), DINOv3-Large model (Siméoni et al., 2025), and SigLIP2-So400m@512 (Tschannen et al., 2025). The same models also provide the global descriptor to generate the respective ranked lists of images per query for re-ranking. The dimensionality after reduction is set to $D = 128$.

We train our model for 10 epochs, sampling triplets of anchors, positives, and negatives as described in (Suma et al., 2024). We use a batch size of 200 triplets and a variable number of local descriptors per image, sampled within the range $[100, 400]$. The network is trained using the AdamW (Loshchilov & Hutter, 2019) optimizer with a learning rate of $5 \cdot 10^{-4}$ and a cosine learning rate schedule with warmup. We follow the same training strategy for both GLDv2 and SOP, with the exception of the learning rate and number of epochs on the SOP dataset, which are set to $1 \cdot 10^{-3}$ and 30 for ELViS, and $5 \cdot 10^{-4}$ and 45 for the compared models, respectively.

For local descriptor extraction, images are resized according to their longer side. We use 770 for DINOv2 and 768 for DINOv3 and SigLIP2 to ensure divisibility by the ViT patch size. The patch tokens output by the backbones are then processed by a local feature detector, which assigns an importance weight to each token. Tokens with the highest weights are selected as local descriptors. Following AMES, we adopt its training strategy and network architecture for the local feature detector, training a different model for each backbone and training set. The dimensionality $D'$ of the output local descriptors is 768 for DINOv2, 1024 for DINOv3, and 1152 for SigLIP2.

It is common practice to ensemble retrieval similarities from global and local descriptors (Zhu et al., 2023; Suma et al., 2024). However, we found this was not necessary for our method to achieve its best performance. In all our experimental results, we report the following for each method: global+local for AMES, R2Former, and local-only for RRT, Chamfer, and ELViS. The ensembling parameter for AMES is tuned on the in-domain validation set, while for R2Former, it is fixed to an equal weight (0.5) for local and global similarity. We do not enforce a consistent setup across all methods, as no single ensembling scheme yields the best results in every setting; instead, we use the ensembling strategy proposed by the original works.

| Method | Alps | | | California | | | Gobi Desert | | | Amazon | | | Toshka Lakes | | |
|---|---|---|---|---|---|---|---|---|---|---|---|---|---|---|---|
| | R@1 | R@10 | R@100 | R@1 | R@10 | R@100 | R@1 | R@10 | R@100 | R@1 | R@10 | R@100 | R@1 | R@10 | R@100 |
| **AnyLoc** | 40.7 | 70.8 | 92.0 | 48.7 | 75.0 | 91.6 | 28.7 | 57.0 | 81.7 | 38.6 | 63.8 | 86.2 | 63.7 | 84.5 | 96.3 |
| **AstroLoc** | 98.1 | 99.5 | 99.8 | 97.4 | 99.2 | 99.8 | 94.6 | 99.2 | 99.9 | 93.0 | 96.9 | 99.1 | 99.0 | 99.6 | 99.9 |
| **No re-ranking** | 35.7 | 65.6 | 91.1 | 50.1 | 78.2 | 93.9 | 26.2 | 53.3 | 84.0 | 31.4 | 61.1 | 83.9 | 52.4 | 78.1 | 94.9 |
| **AMES** | 44.0 | 74.5 | 93.7 | 57.4 | **83.0** | **95.7** | 36.0 | 64.7 | **87.0** | 40.4 | 69.2 | 87.8 | 61.5 | 85.4 | 96.8 |
| **ELViS** | **55.1** | **80.0** | **94.6** | **62.0** | 82.7 | 94.7 | **49.7** | **71.2** | 86.9 | **54.4** | **76.2** | **89.0** | **77.0** | **90.9** | **97.6** |

Table G: **Generalization to extreme domain shifts.** Evaluation on retrieval for Astronaut Photography Localization (APL) (Berton et al., 2024). Performance is measured in Recall@$k$ (R@$k$). APL covers different geographical areas of extreme visual environments. *Top:* reference methods Any-Loc (Keetha et al., 2023), a universal Visual Place Recognition (VPR) approach and the specialist AstroLoc (Berton et al., 2025) trained for the task. *Bottom:* Re-ranking 400 images on top of a DINOv3 backbone. Bold indicates the best performance.

## C  GENERALIZATION TO EXTREME DOMAIN SHIFTS

We evaluate robustness on five test sets of the Astronaut Photography Localization (APL) benchmark (Berton et al., 2024). Although the downstream application is localization, the benchmark is formulated as a standard image retrieval task evaluated via Recall@$k$. The query images are hand-held photographs taken by astronauts, while the database consists of nadir satellite imagery. The goal is to retrieve, for each astronaut photograph query, the corresponding satellite image depicting the same location within the top $k$ ranks. The benchmark includes datasets representing extreme visual environments with distinct scientific importance, such as flood monitoring or disaster response. Table G compares the performance of ELViS and AMES, both using the DINOv3 descriptors and re-ranking 400 images. For reference, we also report performance of two task-specific approaches. ELViS demonstrates high effectiveness compared with the global baseline and AMES.

## D  COMPARISON WITH IMAGE MATCHING METHODS

We compare ELViS against established image matching methods. Specifically, we evaluate Super-Glue (Sarlin et al., 2020) and OmniGlue (Jiang et al., 2024), alongside a spatial verification approach (Philbin et al., 2007). To adapt these matching methods for similarity estimation, we use the count of correspondences and inliers, respectively, as the re-ranking score. This score is ensembled with the global retrieval score via a combination parameter, tuned as in AMES. We consider the landmark datasets as in-domain (ID) for these methods; both SuperGlue and OmniGlue are trained with strong correspondence supervision on landmark datasets, including MegaDepth and the 1M distractor images of $\mathcal{R}$OP+1M.

We evaluate SuperGlue[7] (pre-trained outdoor model) and OmniGlue[8] (SuperPoint + DINOv2 backbone) from their official repositories with the default parameters. To maintain consistency with our pipeline, the number of SuperPoint keypoints is capped at 600. Input images are resized such that the longer side is 1024 pixels, ensuring typically-sized images produce sufficient keypoints while keeping resolutions comparable. For spatial verification, homography is estimated via MAGSAC++ (Barath et al., 2020). Tentative correspondences are obtained by computing optimal transport with dustbins on the descriptor similarity matrix, thresholding these refined similarities, and applying a correspondence reciprocity filtering stage. The similarity threshold, inlier threshold, and reprojection error hyperparameters are tuned on the GLDv2 validation set.

As shown in Table H, SuperGlue is a strong baseline for OOD generalization. It performs competitively with approaches specifically trained for image-to-image similarity (see Table 1). OmniGlue performs poorly, but still improves compared to no re-ranking. Note that both are originally designed for a different task and not for estimating the image-to-image similarity via correspondence counting, as we do in this comparison. Spatial verification improves the initial ranking but remains inferior to Chamfer similarity as the alternative parameter-free option.

---

[7]https://github.com/magicleap/SuperGluePretrainedNetwork
[8]https://github.com/google-research/omniglue

| Method | $\mathcal{R}$OP+1M | GLDv2 | ILIAS | INSTRE | MET | Prod1M | RP2K | SOP-1k | ID | OOD | avg |
|---|---|---|---|---|---|---|---|---|---|---|---|
| No re-ranking | 57.7 | 27.3 | 9.4 | 65.3 | 61.6 | 24.7 | 39.0 | 33.7 | 42.5 | 38.9 | 39.8 |
| Spatial verification | 58.5 | 27.6 | 12.3 | 70.4 | 68.7 | 28.7 | 49.7 | 37.3 | 43.3 | 43.7 | 43.6 |
| SuperGlue | 61.3 | 28.3 | 15.1 | 73.7 | 72.9 | 38.8 | 56.2 | 41.5 | 44.8 | 49.7 | 48.5 |
| OmniGlue | 59.1 | 27.8 | 13.6 | 73.0 | 70.8 | 19.4 | 44.4 | 41.1 | 43.4 | 43.7 | 43.7 |
| ELViS | 68.8 | 32.2 | 18.8 | 80.4 | 77.9 | 41.5 | 59.2 | 52.3 | 50.5 | 55.0 | 53.9 |

Table H: **Comparison with image matching methods**. Landmark datasets are treated as in-domain (ID) for all methods, while the remaining datasets are evaluated as out-of-domain (OOD).

| $\lambda$ | ID | OOD | avg |
|---|---|---|---|
| 0.01 | 46.4 | 47.7 | 47.4 |
| 0.1 | 50.5 | 55.0 | 53.9 |
| 0.5 | 50.7 | 52.1 | 51.7 |
| 1.0 | 48.9 | 48.2 | 48.4 |
| learnable | 51.2 | 54.9 | 54.0 |

| iterations | ID | OOD | avg |
|---|---|---|---|
| 1 | 50.5 | 53.4 | 52.7 |
| 3 | 51.2 | 54.6 | 53.7 |
| 5 | 51.0 | 54.4 | 53.6 |
| 10 | 50.5 | 55.0 | 53.9 |
| 20 | 51.2 | 55.0 | 54.1 |

Table I: **Ablation study of the OT hyperparameters**. Models are trained on the Landmarks (GLDv2) domain. We report average performance over the in-domain (ID), out-of-domain (OOD), and all benchmarks (avg). We assess the impact of the regularization parameter $\lambda$ and the number of OT iterations, varying one hyperparameter at a time while keeping all others at their default values.

## E    IMPACT OF OT HYPERPARAMETERS

Table I reports the performance of ELViS under different values of the regularization parameter $\lambda$ and different numbers of OT iterations. For each configuration, we train a new ELViS model from scratch. The results confirm that increasing the number of OT refinement iterations yields slightly better performance. Conversely, setting $\lambda$ to values either higher or lower than the default value ($\lambda = 0.1$) leads to a substantial performance drop, while making $\lambda$ learnable does not provide any noticeable improvement on average. Using different hyperparameters during training and testing degrades performance considerably and was thus omitted from the results.

## F    DETAILED PERFORMANCE ANALYSIS

In Figure G, we compare the performance on per query basis for ELViS trained in two different domains. What we observe is justified by the train-test domain gaps. In particular, performance on household items is better when training on household items than on landmarks (left plot) and vice versa (middle plot). On INSTRE, which includes a small number of landmarks and household items among a large variety of objects, the best performing model varies a lot across queries.

In Figure H, we show the improvement of ELViS re-ranking over global descriptor retrieval, demonstrated on per-query basis. For the majority of queries, the model improves the ranking, while the cases where it performs poorly are few and marginally impacted.

## G    DETAILED QUANTITATIVE RESULTS

In Tables J and Tables K, we provide the mean and standard deviation across all runs. For every setting, we train three models using a different seed each time. These tables partly repeat information provided in the main paper, but are meant to complement the ones from the main paper with the std and the detailed per dataset performance for DINOv3 and SigLIP.

## H    ADDITIONAL QUALITATIVE EXAMPLES

Figure I demonstrates queries from each test set and the impact of re-ranking by ELViS over a global similarity. In Figure J, we present additional visual examples of correspondences for positive image

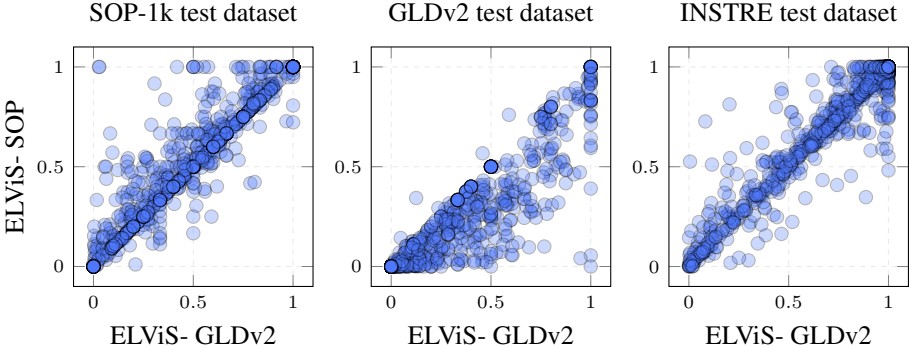

Figure G: **Average precision per query.** Each marker corresponds to the AP of a single query evaluated on different test datasets using DINOv2 as a representation model. x-axis: performance using ELViS trained on SOP dataset. y-axis: performance using ELViS trained on GLDv2 dataset.

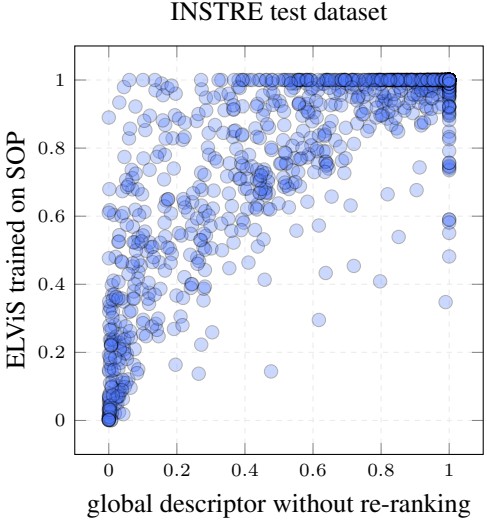

Figure H: **Average precision per query.** Each marker corresponds to the AP of a single query evaluated on INSTRE using DINOv2 as a representation model. y-axis: performance using ELViS trained on SOP dataset. x-axis: performance of the global descriptor without any re-ranking.

pairs from various domains, and in Figure K, the corresponding similarity matrices. Note that an initial similarity matrix with many large values does not necessarily result in many large values after refinement and vote strength transformation. This is due to the optimal transport optimization that jointly processes all similarities and requires some kind of mutual compatibility in the final result.

# I    LLM USAGE IN THIS PAPER

An LLM was used to correct and improve some already written parts of the paper, *i.e.* in the form of an advanced grammar/syntax checker, and for polishing the phrasing. An LLM was never used to generate text from scratch.

| Method | $\mathcal{R}$Oxford | $\mathcal{R}$Paris | GLDv2 | ILIAS | INSTRE | MET | Prod1M | RP2K | SOP-1k | ID | OOD | avg |
|---|---|---|---|---|---|---|---|---|---|---|---|---|
| | | | | | DINOv2 descriptors (Oquab et al., 2024) | | | | | | | |
| **No re-ranking** | 47.3 | 67.9 | 27.3 | 9.4 | 65.3 | 61.6 | 24.7 | 39.0 | 33.7 | 42.5 | 38.9 | 39.8 |
| **Chamfer** | 43.0 | 54.8 | 23.8 | 6.2 | 55.0 | 37.3 | 17.8 | 55.8 | 48.6 | 40.2 | 36.8 | 37.6 |
| **Chamfer+OT** | 55.6 | 65.5 | 23.8 | 14.3 | 76.0 | 74.0 | 35.1 | 55.4 | 46.2 | 42.2 | 50.2 | 48.2 |
| **RRT** | $64.1_{\pm0.6}$ | $74.4_{\pm0.0}$ | $33.1_{\pm0.2}$ | $13.1_{\pm0.6}$ | $72.4_{\pm1.1}$ | $64.1_{\pm0.7}$ | $29.3_{\pm1.7}$ | $60.7_{\pm0.9}$ | $52.1_{\pm0.6}$ | $51.1_{\pm0.3}$ | $48.6_{\pm0.9}$ | $49.2_{\pm0.8}$ |
| $R^2$**Former** | $63.7_{\pm0.2}$ | $73.4_{\pm0.1}$ | $32.6_{\pm0.0}$ | $15.2_{\pm0.1}$ | $77.6_{\pm0.1}$ | $72.0_{\pm0.3}$ | $35.6_{\pm0.3}$ | $47.7_{\pm0.6}$ | $43.7_{\pm0.1}$ | $50.6_{\pm0.1}$ | $48.6_{\pm0.2}$ | $49.1_{\pm0.2}$ |
| **AMES** | $65.6_{\pm0.5}$ | $74.6_{\pm0.1}$ | $34.7_{\pm0.3}$ | $14.6_{\pm0.2}$ | $75.6_{\pm0.3}$ | $70.7_{\pm0.9}$ | $32.3_{\pm2.5}$ | $56.5_{\pm1.1}$ | $48.5_{\pm0.5}$ | $52.4_{\pm0.3}$ | $49.7_{\pm0.9}$ | $50.4_{\pm0.7}$ |
| **ELViS** | $64.3_{\pm0.3}$ | $73.3_{\pm0.2}$ | $32.2_{\pm0.0}$ | $18.8_{\pm0.1}$ | $80.4_{\pm0.2}$ | $76.5_{\pm0.4}$ | $41.5_{\pm0.3}$ | $59.2_{\pm1.0}$ | $52.3_{\pm0.2}$ | $50.5_{\pm0.1}$ | $55.0_{\pm0.2}$ | $53.9_{\pm0.3}$ |
| | | | | | DINOv3 descriptors (Siméoni et al., 2025) | | | | | | | |
| **No re-ranking** | 59.7 | 75.4 | 31.7 | 26.5 | 88.2 | 77.2 | 64.9 | 61.6 | 44.5 | 49.6 | 60.5 | 57.8 |
| **Chamfer** | 51.0 | 69.9 | 24.2 | 6.5 | 71.3 | 48.6 | 34.1 | 62.7 | 43.8 | 42.3 | 44.5 | 44.0 |
| **Chamfer+OT** | 56.1 | 66.9 | 21.9 | 12.3 | 86.7 | 63.0 | 48.8 | 57.4 | 40.5 | 41.7 | 51.4 | 49.0 |
| **RRT** | $72.7_{\pm0.9}$ | $82.0_{\pm0.3}$ | $36.9_{\pm0.1}$ | $25.8_{\pm1.8}$ | $84.0_{\pm1.5}$ | $58.3_{\pm3.5}$ | $50.6_{\pm2.9}$ | $62.0_{\pm1.8}$ | $56.1_{\pm0.3}$ | $57.1_{\pm0.3}$ | $56.2_{\pm2.0}$ | $56.4_{\pm1.6}$ |
| $R^2$**Former** | $70.8_{\pm0.2}$ | $80.1_{\pm0.4}$ | $36.5_{\pm0.1}$ | $34.3_{\pm0.3}$ | $92.3_{\pm0.1}$ | $79.3_{\pm0.5}$ | $62.9_{\pm0.4}$ | $63.0_{\pm0.1}$ | $51.4_{\pm0.1}$ | $56.0_{\pm0.2}$ | $63.9_{\pm0.3}$ | $61.9_{\pm0.2}$ |
| **AMES** | $75.1_{\pm0.7}$ | $82.2_{\pm0.3}$ | $38.6_{\pm0.2}$ | $32.4_{\pm0.8}$ | $89.6_{\pm0.1}$ | $72.8_{\pm0.4}$ | $61.1_{\pm0.1}$ | $65.3_{\pm0.1}$ | $55.7_{\pm0.9}$ | $58.6_{\pm0.2}$ | $62.8_{\pm0.5}$ | $61.8_{\pm0.4}$ |
| **ELViS** | $74.0_{\pm0.2}$ | $79.9_{\pm0.0}$ | $37.0_{\pm0.0}$ | $41.2_{\pm0.0}$ | $93.4_{\pm0.1}$ | $81.0_{\pm0.2}$ | $63.9_{\pm0.1}$ | $68.1_{\pm0.1}$ | $57.0_{\pm0.2}$ | $56.9_{\pm0.1}$ | $67.4_{\pm0.1}$ | $64.8_{\pm0.1}$ |
| | | | | | SigLIP2 descriptors (Tschannen et al., 2025) | | | | | | | |
| **No re-ranking** | 17.3 | 47.8 | 18.3 | 22.4 | 89.8 | 61.7 | 70.2 | 40.6 | 62.2 | 25.4 | 57.8 | 49.7 |
| **Chamfer** | 23.0 | 59.1 | 24.6 | 22.9 | 83.7 | 57.3 | 67.1 | 52.9 | 68.3 | 32.8 | 58.7 | 52.2 |
| **Chamfer+OT** | 23.0 | 54.9 | 19.7 | 28.3 | 90.0 | 69.0 | 69.3 | 50.2 | 67.3 | 29.3 | 62.3 | 54.1 |
| **RRT** | $28.3_{\pm0.5}$ | $62.0_{\pm0.0}$ | $29.8_{\pm0.1}$ | $24.9_{\pm2.5}$ | $82.9_{\pm1.6}$ | $59.2_{\pm5.0}$ | $65.3_{\pm2.8}$ | $48.4_{\pm2.0}$ | $69.7_{\pm0.7}$ | $37.5_{\pm0.1}$ | $58.4_{\pm2.5}$ | $53.2_{\pm1.9}$ |
| $R^2$**Former** | $25.7_{\pm0.4}$ | $59.8_{\pm0.1}$ | $27.7_{\pm0.4}$ | $31.3_{\pm0.1}$ | $90.4_{\pm1.3}$ | $69.4_{\pm1.0}$ | $70.5_{\pm1.3}$ | $47.8_{\pm1.5}$ | $68.4_{\pm1.1}$ | $35.2_{\pm0.3}$ | $63.0_{\pm1.1}$ | $56.0_{\pm1.0}$ |
| **AMES** | $27.7_{\pm1.2}$ | $61.1_{\pm0.7}$ | $29.9_{\pm0.4}$ | $30.4_{\pm1.2}$ | $89.0_{\pm1.0}$ | $65.9_{\pm0.5}$ | $72.5_{\pm0.6}$ | $48.2_{\pm1.0}$ | $70.1_{\pm1.6}$ | $37.1_{\pm0.7}$ | $62.7_{\pm1.0}$ | $56.3_{\pm0.9}$ |
| **ELViS** | $27.8_{\pm0.0}$ | $61.5_{\pm0.1}$ | $28.1_{\pm0.0}$ | $41.3_{\pm0.2}$ | $93.5_{\pm0.1}$ | $76.2_{\pm0.4}$ | $73.4_{\pm0.1}$ | $55.9_{\pm0.2}$ | $72.1_{\pm0.0}$ | $36.4_{\pm0.1}$ | $68.7_{\pm0.1}$ | $60.6_{\pm0.1}$ |

Table J: **mAP mean and standard deviation for training on landmarks (GLDv2).** Three different backbones used as a representation model.

| Method | $\mathcal{R}$Oxford | $\mathcal{R}$Paris | GLDv2 | ILIAS | INSTRE | MET | Prod1M | RP2K | SOP-1k | ID | OOD | avg |
|---|---|---|---|---|---|---|---|---|---|---|---|---|
| **No re-ranking** | 47.3 | 67.9 | 27.3 | 9.4 | 65.3 | 61.6 | 24.7 | 39.0 | 33.7 | 33.7 | 40.7 | 39.8 |
| **Chamfer** | 29.7 | 62.6 | 15.4 | 6.7 | 63.2 | 45.2 | 24.7 | 50.3 | 50.2 | 50.2 | 35.9 | 37.7 |
| **Chamfer+OT** | 43.7 | 60.8 | 18.6 | 11.7 | 75.9 | 71.6 | 37.5 | 52.7 | 45.8 | 45.8 | 45.7 | 45.7 |
| **RRT** | $28.2_{\pm1.7}$ | $58.7_{\pm1.3}$ | $10.8_{\pm1.1}$ | $12.2_{\pm1.5}$ | $68.4_{\pm1.8}$ | $25.5_{\pm2.8}$ | $32.2_{\pm1.8}$ | $46.9_{\pm3.9}$ | $57.1_{\pm0.3}$ | $57.1_{\pm0.3}$ | $34.2_{\pm2.0}$ | $37.1_{\pm1.5}$ |
| $R^2$**Former** | $43.4_{\pm1.0}$ | $67.5_{\pm0.3}$ | $23.7_{\pm0.5}$ | $12.9_{\pm0.3}$ | $73.4_{\pm0.4}$ | $59.3_{\pm0.7}$ | $32.8_{\pm1.9}$ | $42.0_{\pm0.7}$ | $51.1_{\pm0.3}$ | $51.1_{\pm0.3}$ | $42.8_{\pm0.7}$ | $43.8_{\pm0.7}$ |
| **AMES** | $46.2_{\pm1.0}$ | $65.1_{\pm1.5}$ | $17.2_{\pm1.1}$ | $12.4_{\pm0.5}$ | $72.2_{\pm1.0}$ | $44.2_{\pm1.8}$ | $36.7_{\pm1.9}$ | $51.8_{\pm2.5}$ | $56.7_{\pm0.6}$ | $56.7_{\pm0.6}$ | $41.4_{\pm1.5}$ | $43.3_{\pm1.2}$ |
| **ELViS** | $50.5_{\pm0.6}$ | $68.9_{\pm0.1}$ | $22.9_{\pm0.3}$ | $18.6_{\pm0.0}$ | $81.1_{\pm0.2}$ | $76.7_{\pm0.3}$ | $44.1_{\pm0.1}$ | $54.2_{\pm0.9}$ | $54.9_{\pm0.1}$ | $54.9_{\pm0.1}$ | $51.0_{\pm0.3}$ | $51.5_{\pm0.2}$ |

Table K: **mAP mean and standard deviation for training on household items (SOP).** DINOv2 used as a representation model.

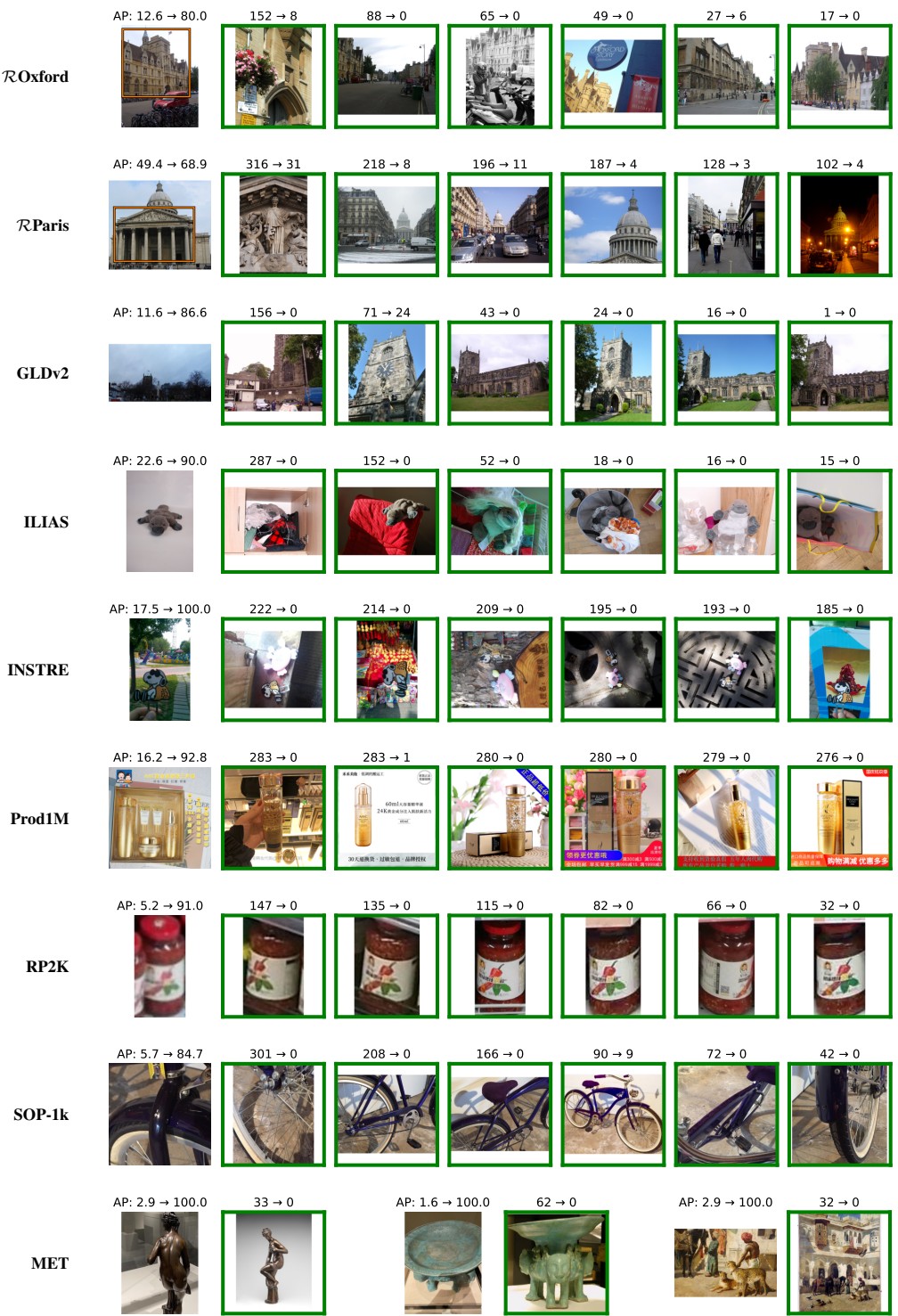

Figure I: **The impact of re-ranking with ELViS.** We show a single query and its database positives per row for each test set, with the exception of the MET dataset where three queries are shown. The text above each query denotes the change in Average Precision. The text above the positive images denotes the number of negative images ranked before the positive using only global similarity (1) and after re-ranking with ELViS (2), denoted as (1)→(2). The positives are ordered based on the difference (1)-(2) in descending order.

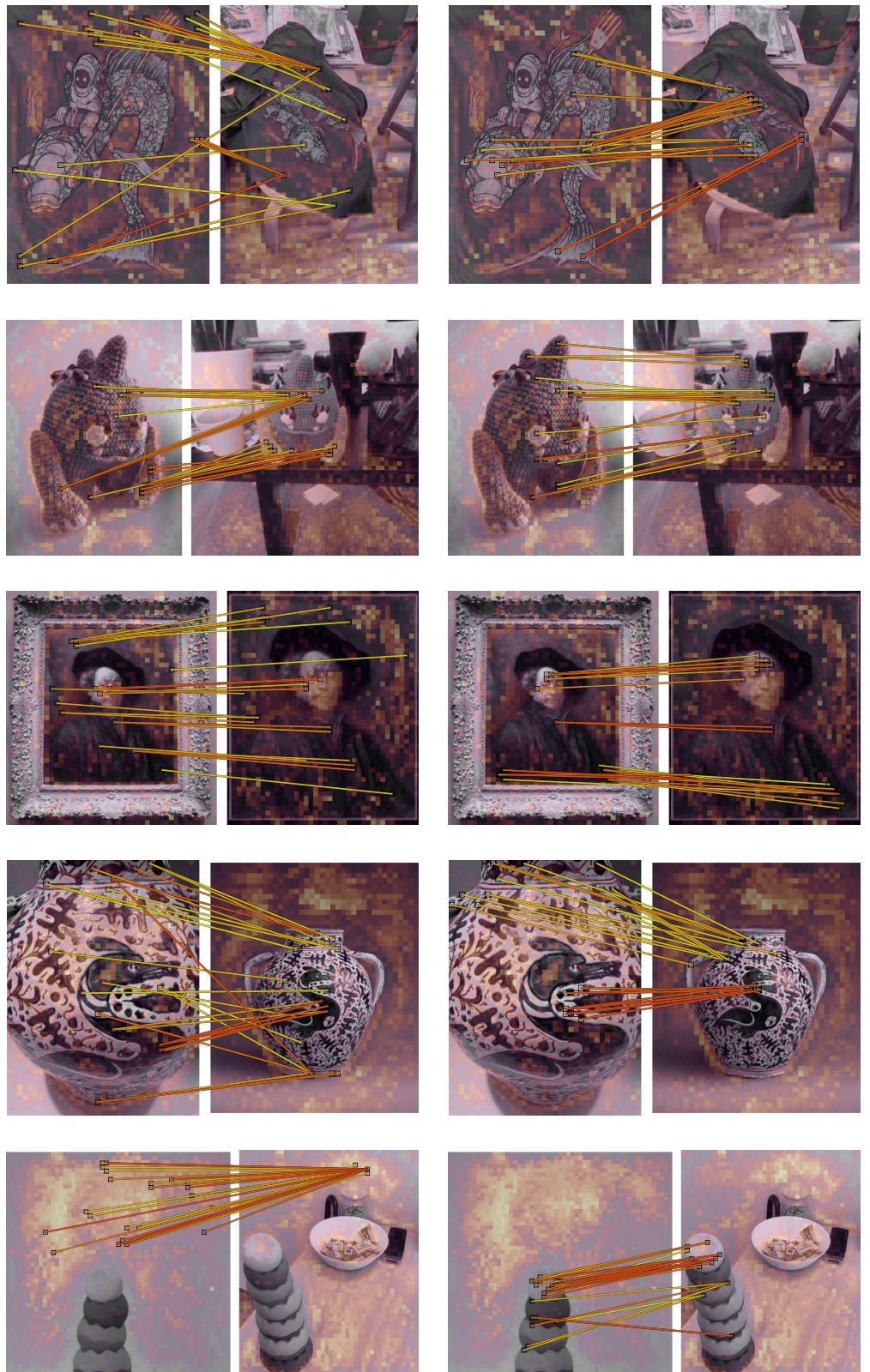

Figure J: **Visualization of strong correspondences** before (left) and after (right) refinement with optimal transport.

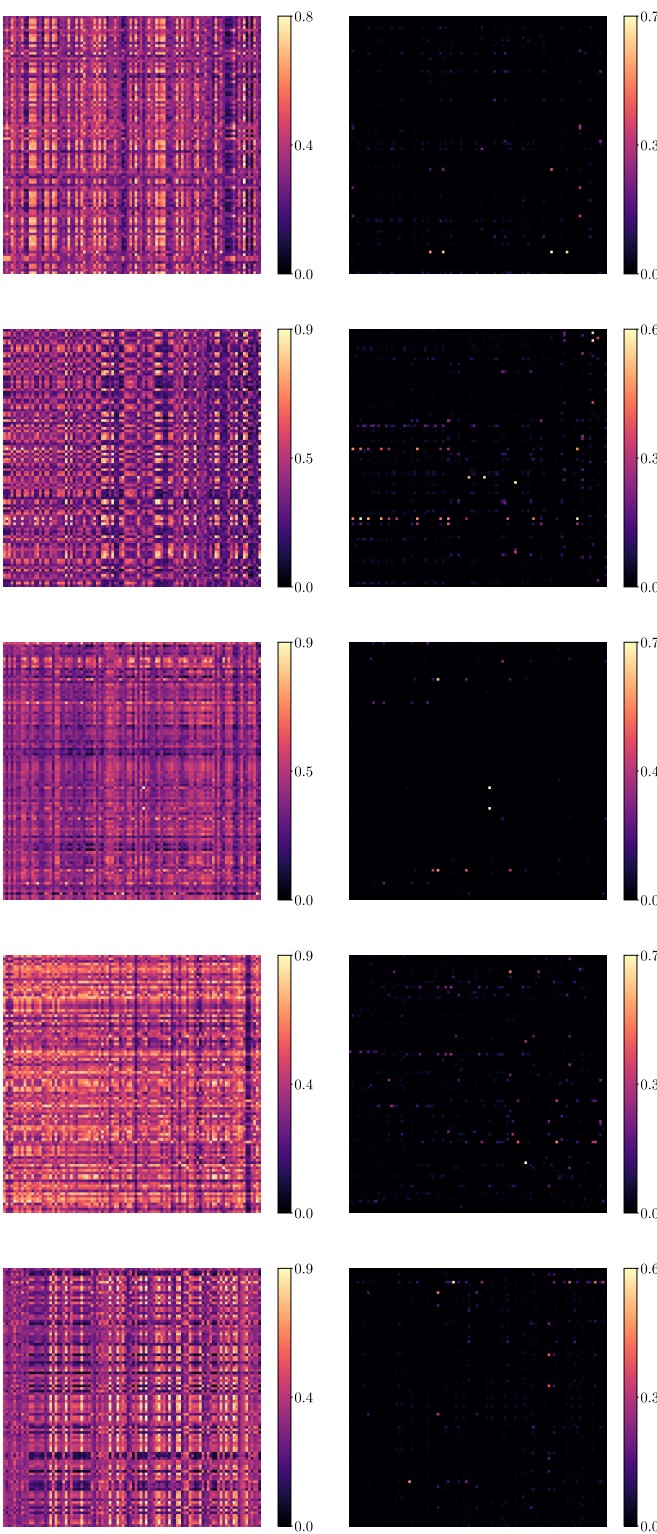

Figure K: **Visualization of similarity matrices** before (left) and after (right) refinement with optimal transport with individual values passed through function $f$. A subset of 100 descriptors is used. These examples correspond to the ones of Figure J.

