# OpenReview forum: "ELViS: Efficient Visual Similarity from Local Descriptors that Generalizes Across Domains"
_ICLR.cc/2026/Conference — ICLR 2026 Poster_

### Official Review · Reviewer_2TKs · 2025-10-22

**Soundness:** 2
**Presentation:** 3
**Contribution:** 2
**Rating:** 6
**Confidence:** 3

**Summary:**

This paper introduces ELViS (Efficient Local Visual Similarity), a novel model for instance-level image retrieval. The core problem addressed is single-source domain generalization: training a model on one domain (e.g., landmarks) and having it perform well on retrieval tasks in unseen domains.

**Method Overview:**
- Input: Local descriptors are extracted from two images using a foundation model (like DINOv2).
 - Similarity Matrix Refinement: The local descriptor similarity matrix is refined using Optimal Transport with a key innovation: descriptor-dependent dustbin gains. This allows the model to learn to ignore uninformative descriptors (e.g., from the background).
- Vote Aggregation: For each descriptor, the strongest correspondence (similarity) is selected as a `"vote." A small, learned function \$f\$ transforms these vote strengths, and they are summed to produce a final, global image similarity score.
- Training: The model uses a modified Binary Cross-Entropy loss with a second learned function g to reshape the penalty curve during training, which is discarded at inference.


**Main Results:**
The authors demonstrate that ELViS achieves state-of-the-art performance on a comprehensive benchmark of 8 datasets, showing superior generalization to out-of-domain data while being significantly faster and more parameter-efficient than competing transformer-based methods (RRT, R²Former, AMES).

**Strengths:**

- The paper is well written and clear to follow.
- SOTA Cross-Domain Generalization: The primary strength of ELViS is its ability to perform robustly on domains unseen during training. It consistently outperforms all competitors on out-of-domain (OOD) datasets.
- The method is efficient on both parameters count and latency aspects.
- Ablation studies on the contribution of each component is provided and well demonstrated.

**Weaknesses:**

- Justification for \$g\$: The use of the learned function \$g\$ in the loss, which is discarded at inference, is an unconventional and somewhat non-standard technique. While it works well empirically, a more rigorous theoretical explanation for why this is necessary and why discarding it is valid would strengthen the method.
- Performance on In-Domain Data: In some in-domain (ID) settings, ELViS is outperformed by other methods. This suggests that while its bias towards generalization is powerful, it might come at the cost for optimal performance on the training domain itself.
- Novelty: The method primarily relies on existing components, which somewhat limits its degree of novelty.

**Questions:**

-

---

> ### Author Response · Authors · 2025-11-20
>
> We thank the reviewer for their constructive comments. Here, we are providing detailed responses to all reviewer feedback, and we will submit an updated manuscript next week to incorporate the current and forthcoming comments.
> ___
> > **Justification for $g$: The use of the learned function $g$ in the loss, which is discarded at inference, is an unconventional and somewhat non-standard technique. While it works well empirically, a more rigorous theoretical explanation for why this is necessary and why discarding it is valid would strengthen the method.**
>
> We agree that it is non-standard and we believe it is part of the novelty of our work. Such an expendable module bears similarities to two existing cases:
> * Some processing blocks are applied on representation vectors/tensors during training and are removed during testing. This has been standard in SSL (SimCLR [A] and many other approaches after that including [B,C]), to encourage generalization to other tasks (which is similar to our case even though we target domain generalization). A similar case is used in deep metric learning [D]. These are applied on whole latent representations, while ours on the scalar similarity.
> * A learnable temperature that multiplies the scalar confidence used in cross-entropy loss (e.g. as in CLIP [E]). This learnable parameter affects the loss landscape during training but does not affect the arg-max prediction during testing. We also apply $g$ on the scalar similarity but our transformation is non-linear.
>
> Thank you, the updated manuscript will integrate the above discussion.
>
> References: \
> [A] Chen et al. ”A simple framework for contrastive learning of visual representations.” ICML, 2020. \
> [B] Zbontar et al. "Barlow twins: Self-supervised learning via redundancy reduction." ICML 2021. \
> [C] Sariyildiz et al.  "No reason for no supervision: Improved generalization in supervised models." ICLR 2023. \
> [D] Seidenschwarz et al. “Learning Intra-Batch Connections for Deep Metric Learning.” ICML 2021. \
> [E] Radford et al. "Learning transferable visual models from natural language supervision." ICML 2021.
>
> ___
> > **Performance on In-Domain Data: In some in-domain (ID) settings, ELViS is outperformed by other methods. This suggests that while its bias towards generalization is powerful, it might come at the cost for optimal performance on the training domain itself.**
>
> This is true. ELViS excels in unseen domains, but is weaker in the seen domain. This is valuable because in open-world settings and many retrieval applications one will be testing on unseen object categories / domains. Such observation is in support of our message that similarity-based processing generalizes better than feature-based processing, while feature-based processing is stronger in the seen domain. We believe this is a valuable take-home message.
>
> To further support this point, we conducted an additional experiment where we created a “hybrid” model combining AMES and ELViS. Specifically, we first feed the two descriptor sets through the same sequence of transformer blocks used in AMES, and then treat the resulting output tokens as a refined descriptor set, which we pass to ELViS. We train this hybrid model end-to-end.
>
> The average, ID, and OOD performance is 54.0, 52.1, and 54.7, respectively. This is interesting because such feature-based processing made ELViS perform better in ID, and compromised the OOD performance slightly.
>
> ___
> > **Novelty: The method primarily relies on existing components, which somewhat limits its degree of novelty.**
>
> While built on existing components, the overall design is new, achieving strong generalization, efficiency, and interpretability, and delivering a clear message on the value of inductive biases over black-box architectures. Furthermore, some of our design choices bring added novelty: a) our improved optimal transport variant with learnable gains and a learnable voting mechanism,  b)  the processing of scalar local descriptor similarities through an MLP, c) the expendable function that improves training.

---

### Official Review · Reviewer_UYEL · 2025-10-28

**Soundness:** 2
**Presentation:** 3
**Contribution:** 3
**Rating:** 6
**Confidence:** 4

**Summary:**

This paper introduced a image-to-image similarity model that promotes cross-domain transfer, namely Efficient Local Visual Similarity (ELViS). In order to facilitate faster and more explainable image retrieve in cross-domain scenarios, ELViS leveraged optimal transportation to refine the local description similarity matrix S’, and then aggregate a learnable voting process to transfer the local similarity to the global similarity for the further image retrieval.
Their major technical contribution of their work lies in (1) the construction of refined local-description similarity matrix S’, which contains traditional similarity matrix S, data-dependent gains (learned by parametric method) that suppress uninformative descriptors (dustbins) and the learnable scalars that stands for transportation mass for dustbins. (2) the voting mechanism that transfer local descriptions to global description. (3) The authors built a benchmarking protocol that unified 8 existing datasets across various domains.

**Strengths:**

(1) The construction of refined local description similarity matrix is novel and intuitive. Especially the introduction of dustbin that avoids the hard comparison between different image instance. And the implementation of optimal transportation to refine the similarity matrix is intuitive and reasonable. Additionally, compared to the former strategy, this paper used the parametric method to empower the model with the ability to adjust the dustbin with self-supervision, making the model more interpretative and flexible.
(2) They introduced 8 benchmarks that contains various kinds of domains, and they are the first work to conduct such an extensive evaluation of single-source domain generalization in instance-level retrieval. And the model performed well on those datasets, especially on out-of-domain retrieval scenarios.
(3) Most of the figures and narrations in this paper (except for Introduction) is good and logical, stating their motivation and insights.
(4) The experiments and further analysis is abundant and showed the model’s effectiveness.

**Weaknesses:**

(1) Compared to their technical contribution, their narration in the introduction is less satisfying and cannot specify their motivations.
a. The authors should detailed the reason why the focus on local descriptors is better than the global descriptors for the cross-domain image retrieval. Even though the author mentioned the intepretability and time cost, they can delve deeper into the explanation in representation space, making their statement and motivation less trivial and more solid.
b. It would be better for the authors to add a figure for this comparison, making their motivation more intuitive.
c. It would be better for the author to itemize and highlight their contribution in end of the introduction, making their contribution more clear for the readers.
(2) The authors should simplify the Section.2 (Related Work), instead explicitly detail their motivation, contribution and analysis on previous researches in the introduction.
(3) Even though model’s in-domain performances didin’t achieve state-of-the-art performances, it is acceptable.

**Questions:**

See weakness

---

> ### Author Response · Authors · 2025-11-20
>
> We thank the reviewer for their constructive comments. Here, we are providing detailed responses to all reviewer feedback, and we will submit an updated manuscript next week to incorporate the current and forthcoming comments.
> ___
> > **a. The authors should detailed the reason why the focus on local descriptors is better than the global descriptors for the cross-domain image retrieval.**
>
> Both global and local descriptors are derived from frozen foundation models. Thanks to their large-scale pre-training, these descriptors are well-suited for open-world retrieval. Global descriptors can directly provide similarity scores (e.g., cosine similarity) without additional training. In contrast, estimating image-to-image similarity from sets of local descriptors requires learning. We ensure that this learning process and the resulting model generalize effectively.
> Re-ranking with local descriptors is a common strategy to improve retrieval performance in cases involving small objects, occlusions, or cluttered scenes. By focusing on local descriptors we are allowed to learn a model whose parameters operate directly in the space of _descriptor similarities_, rather than in the space of the descriptors themselves (as done, for example, in AMES). This distinction is a key factor behind the strong out-of-domain performance we observe.
>
> We would also like to clarify terminology: our work addresses cross-domain generalization, not to be confused with cross-domain retrieval. Cross-domain generalization refers to training the retrieval model on one domain and evaluating it on a different domain, where both the query and database images come from this new domain. Cross-domain retrieval, in contrast, typically refers to scenarios where the query and database images originate from different domains.
> Thank you again for the comment. We will clarify all of the above in the updated manuscript (currently only partially addressed in lines 050–053).
> ___
> > **b. It would be better for the authors to add a figure for this comparison, making their motivation more intuitive.**
>
> We are unsure about what kind of figure the reviewer has in mind, but we will be glad to hear more about it and try to improve the manuscript.
> ___
> > **c. It would be better for the author to itemize and highlight their contribution in end of the introduction, making their contribution more clear for the readers.**
>
> Thank you, we will restate our contributions in a more clear way:
>
> We introduce ELVIS, a novel _similarity-based_ reranking model that \
> a) operates directly on sets of local-descriptor similarities via a novel OT formulation, \
> b) is composed of simple, lightweight components, and \
> c) provides a high degree of interpretability at multiple stages of the pipeline.
>
> We evaluate ELVIS on eight diverse instance-level benchmarks and show that, in addition to being substantially faster, it delivers large performance gains on out-of-domain datasets while matching the performance of much heavier models on the training domain.
> ___
> > **The authors should simplify the Section.2 (Related Work), instead explicitly detail their motivation, contribution and analysis on previous researches in the introduction.**
>
> Thank you for the comment. We understand that the concern refers to the final paragraphs of each subsection, and we agree with the reviewer. In the revised manuscript, we will remove these parts from their current locations and integrate the corresponding content into the introduction, unless it is already covered elsewhere.
> ___
> > **Even though model’s in-domain performances didin’t achieve state-of-the-art performances, it is acceptable.**
>
> This is true. ELViS excels in unseen domains, but is weaker in the seen domain. This is valuable because in open-world settings and many retrieval applications one will be testing on unseen object categories / domains. Such observation is in support of our message that similarity-based processing generalizes better than feature-based processing, while feature-based processing is stronger in the seen domain. We believe this is a valuable take-home message.
>
> To further support this point, we conducted an additional experiment where we created a “hybrid” model combining AMES and ELViS. Specifically, we first feed the two descriptor sets through the same sequence of transformer blocks used in AMES, and then treat the resulting output tokens as a refined descriptor set, which we pass to ELViS. We train this hybrid model end-to-end.
>
> The average, ID, and OOD performance is 54.0, 52.1, and 54.7, respectively. This is interesting because such feature-based processing made ELViS perform better in ID, and compromised the OOD performance slightly.

---

### Official Review · Reviewer_dvx4 · 2025-10-29

**Soundness:** 4
**Presentation:** 3
**Contribution:** 3
**Rating:** 6
**Confidence:** 4

**Summary:**

The paper presents ELViS, a re-ranking technique which is robust across domains. Elvis can be applied on top of common vision foundation models like DINOv2, DINOv3 and SigLIP2, and uses the transformer's local features by applying a lightweight post-processing to output a similarity score between two images, which is then used to re-rank a shortlist of retrieved candidates.

**Strengths:**

The paper outlines, demonstrates, and tackles a clear problem, which is that re-ranking methods trained on one domain underperform on others.
Elvis shows overall improvement when used on OOD data.
The paper is well written

**Weaknesses:**

1. Can Elvis work when images are of different resolutions? Given that f is an MLP it seems like Elvis would require images at a fixed resolution.

2. I don't fully understand Figure 3. A better caption would be helpful

3. Splitting results on ROxford and RParis would make results clearer and more comparable with other literature. Also which sets of ROxford and RParis are used (easy, medium, hard)?

4. The dataset table should report also the sizes of train/val/test sets for the two datasets used for training

5. A couple of images per dataset would help the reader to understand the domain gap between any two datasets

6. I believe the retrieval is performed with the same model of which the local features are used? I don't see this explicitly stated in the paper

7. Most importantly, comparing with image matching methods would be really helpful for the reader. Are the presented methods relevant, or should methods like SuperGlue be used for re-ranking in these domains?

**Questions:**

See the weaknesses stated above

---

> ### Author Response · Authors · 2025-11-20
>
> We thank the reviewer for their constructive comments. Here, we are providing detailed responses to all reviewer feedback, and we will submit an updated manuscript next week to incorporate the current and forthcoming comments.
>
> ___
> > **1. Can Elvis work when images are of different resolutions?**
>
> It can. The function $f$ is applied to each __individual__ local-descriptor (scalar) similarity independently. It does not depend on the number of descriptors or the size of the similarity matrix. Our MLP is different from MLPs used in some deep architectures where a full 3D feature tensor is flattened and fed into the MLP. In contrast, ELViS operates purely at the level of local-descriptor similarities, making it agnostic to image resolution or descriptor count. As a result, we can process images of any resolution: we first extract local features, and ELViS operates directly on the resulting descriptor sets.
>
> ___
> > **2. I don't fully understand Figure 3. A better caption would be helpful**
>
> In Figure 3, we illustrate the output of the learned MLP over the range of possible input values. Both MLPs corresponding to functions $f$ and $g$ take a _scalar_ as input and produce a _scalar_ output. For the visualization, we sample input values from our dataset of positive and negative image pairs (local descriptor similarities for $f$, and global similarities for $g$) and plot the resulting distributions. We thank the reviewer for the suggestion, and we will update the figure caption to clarify this.
>
> ___
> > **3. Splitting results on ROxford and RParis would make results clearer and more comparable with other literature. Also which sets of ROxford and RParis are used (easy, medium, hard)?**
>
> We thank the reviewer for the suggestion. We will make sure to also report results on ROxford and RParis separately. In all cases we use the Medium + Hard sets, a common choice in related works.
>
> ___
> > **4. The dataset table should report also the sizes of train/val/test sets for the two datasets used for training**
>
> Thank you, we will add the sizes.
> ___
> > **5. A couple of images per dataset would help the reader to understand the domain gap between any two datasets**
>
> Thank you, this is a good suggestion. We will add a figure with those in the supplementary material. The updated manuscript will include those.
> ___
> > **6. I believe the retrieval is performed with the same model of which the local features are used? I don't see this explicitly stated in the paper**
>
> This is correct. Whenever we used SigLIP for local descriptors, we also use it for global descriptor too. The same for DINOv3. We will clarify in the updated manuscript.
> ___
> > **7. Comparing with image matching methods [....] SuperGlue used for re-ranking in these domains?**
>
> This is a good suggestion. We are currently evaluating pre-trained SuperGlue (SuperPoint features) for re-ranking on top of DINOv2 global retrieval. We already have the results for some datasets, reported in the table below. It provides improvement on all datasets compared to global retrieval only. Notably, it outperforms RRT, R2Former, and AMES on Prod1M and MET. Nevertheless, ELViS consistently performs better on all datasets. Our experiment is ongoing and we will report the results over all datasets once we obtain them and we will integrate in the updated manuscript.
>
> | Method | | GLDv2 | INSTRE | MET | Prod1M | RP2K | SOP-1k |
> | :--- | :---: | :---: | :---: | :---: | :---: | :---: | :---: |
> | **No re-ranking** | | 27.3 | 65.3 | 61.5 | 24.7 | 39.0 | 33.7 |
> | **RRT** | | 33.1 | 72.4 | 64.1 | 29.3 | 60.7 | 52.1 |
> | **R2Former** | | 32.6 | 77.6 | 72.0 | 35.6 | 47.7 | 43.7 |
> | **AMES** | | 34.7 | 75.6 | 70.7 | 32.3 | 56.5 | 48.5 |
> | **SuperGlue** | | 28.3 | 73.7 | 72.9 | 38.8 | 56.2 | 41.5 |
> | **ELViS** | | 32.2 | 80.4 | 77.9 | 41.5 | 59.2 | 52.3 |

---

> > ### Comment · Reviewer_dvx4 · 2025-11-24
> > **Response to rebuttal**
> >
> > The rebuttal clears my concerns. I trust that the authors will update the paper with comparison with image matching methods.

---

> > > ### Author Response · Authors · 2025-11-28
> > >
> > > We thank the reviewer for their time and positive feedback.
> > > ___
> > > > **7. Comparing with image matching methods [....] SuperGlue used for re-ranking in these domains?**
> > >
> > > We present the full results for SuperGlue below. ELViS outperforms SuperGlue across all datasets, with the exception of ROP+1M, where performance is comparable, while SuperGlue is outperformed by AMES. Note that, SuperGlue is trained on landmark images including correspondence supervision, while all methods in our work are trained with image-level supervision. Moreover, SuperGlue's training set includes the 1M distractor images of ROP, making evaluation on this dataset unfair. These results will be integrated in the updated manuscript.
> > >
> > > | Method | | ROP+1M | GLDv2 | ILIAS | INSTRE | MET | Prod1M | RP2K | SOP-1k |
> > > | :--- | :---: | :---: | :---: | :---: | :---: | :---: | :---: | :---: | :---: |
> > > | **No re-ranking** | | 57.7 | 27.3 | 9.3 | 65.3 | 61.5 | 24.7 | 39.0 | 33.7 |
> > > | **RRT** | | 69.2 | 33.1 | 13.1 | 72.4 | 64.1 | 29.3 | 60.7 | 52.1 |
> > > | **R2Former** | | 68.5 | 32.6 | 15.2 | 77.6 | 72.0 | 35.6 | 47.7 | 43.7 |
> > > | **AMES** | | 70.1 | 34.7 | 14.6 | 75.6 | 70.7 | 32.3 | 56.5 | 48.5 |
> > > | **SuperGlue** | | 69.2 | 28.3 | 15.1 | 73.7 | 72.9 | 38.8 | 56.2 | 41.5 |
> > > | **ELViS** | | 68.8 | 32.2 | 18.8 | 80.4 | 77.9 | 41.5 | 59.2 | 52.3 |

---

### Official Review · Reviewer_6e19 · 2025-10-31

**Soundness:** 3
**Presentation:** 3
**Contribution:** 3
**Rating:** 6
**Confidence:** 5

**Summary:**

Instance-level image retrieval struggles with poor cross-domain generalization—existing models overfit to domain-specific data and fail on unseen domains, due to scarce cross-domain training data and reliance on representation space.
Specifically,
1.Refine the local descriptor similarity matrix using entropy-regularized optimal transport (OT) with descriptor-dependent dustbin gains to filter uninformative patches
2.Aggregate global similarity: select strongest local similarities per descriptor, weight them via a learnable function f, and sum—trained with modified BCE loss.

**Strengths:**

The paper shifts instance-level image retrieval from representation space to similarity space for stronger cross-domain robustness, refines optimal transport (OT) with descriptor-dependent dustbin gains , and creates the first unified benchmark for single-source cross-domain retrieval—uniting 8 datasets across 5 domains to standardize generalization evaluation.

**Weaknesses:**

Chowdhury (2022) also employed optimal transport (OT) to address the instance-level image retrieval task. Therefore, I have concerns about the technical innovation of this paper.
•	Optimal Transport (OT) Parameters: The paper uses 10 iterations of the Sinkhorn-Knopp algorithm and a regularization term \(\lambda = 0.1\), but it does not explain: i) Why 10 iterations (not 5 or 20)? ii) Why \(\lambda = 0.1\)? Cuturi (2013) shows \(\lambda\) directly impacts OT’s accuracy-efficiency tradeoff
•	Extreme Domains: The paper does not evaluate on domains with radical visual differences from natural images (e.g., infrared images, underwater photos, remote sensing imagery)—scenarios where cross-domain retrieval is highly valuable (e.g., satellite image matching for disaster response).
•	Small-Sample Training: The paper uses large training sets (GLDv2 has 762K images, SOP has 60.5K), but many real-world domains have only 100–1000 labeled samples. It is unknown if ELViS’s small parameter count (96K) translates to good small-sample performance.


Cuturi, M. (2013). Sinkhorn distances: Lightspeed computation of optimal transport. Advances in neural information processing systems, 26.
Chowdhury, P. N., Bhunia, A. K., Gajjala, V. R., Sain, A., Xiang, T., & Song, Y. Z. (2022). Partially does it: Towards scene-level fg-sbir with partial input. In Proceedings of the IEEE/CVF Conference on Computer Vision and Pattern Recognition (pp. 2395-2405).

**Questions:**

For the questions , please refer to the above summary of weaknesses.

---

> ### Author Response · Authors · 2025-11-20
>
> We thank the reviewer for their constructive comments. Here, we are providing detailed responses to all reviewer feedback, and we will submit an updated manuscript next week to incorporate the current and forthcoming comments.
> ___
> > **Chowdhury (2022) also employed optimal transport (OT) to address the instance-level image retrieval task.**
>
> Even though this work is also using OT for a retrieval task, there are several key differences between the two methods:
> * We use dustbins and propose the descriptor-dependent gains, which is new and makes the method effective
> * Use of the OT result: we process with chamfer matching and a learnable counting process which is new, while Chowdhury et al. simply use the result as weights in a weighted aggregation of all the region-level similarities.
> * To obtain differentiability: we use entropy regularization allowing to solve with the SK algorithm, while they rely on Lagrange multipliers and KKT conditions.
> * Our application is cross-domain generalization (train on X, test on Y), while theirs is cross-domain retrieval (query with X to retrieve Y).
> * Note that our set of contributions building on OT results in a re-ranking model that is the fastest among existing, with the best generalization ability.
>
> ___
> > **Optimal Transport (OT) Parameters: i) Why 10 iterations (not 5 or 20)? ii) Why (\lambda = 0.1)?**
>
> We chose 10 iterations as a trade-off that balances the model speed and performance (mentioned also in Appendix B). Sinkhorn-Knopp algorithm is the computational bottleneck of ELViS where inference time grows almost linearly with the number of iterations. For $\lambda$, we empirically observe the value of 0.1 to perform well and be robust to different number of iterations.
>
> To further support the chosen parameters and to study the model’s sensitivity we further conduct a series of new experiments.
>
> Changing the value of iterations and $\lambda$ during test-time using our default model (trained with 10 iterations and $\lambda$):
> * As a reference: the default model which is trained and tested for 10 iterations and $\lambda=0.1$ achieves average performance of 53.9.
> * Testing for 3, 5, 20 iterations achieves average performance of 51.9 ,53.0, and 54.2, respectively. Therefore, more iterations bring a slight improvement but compromise inference speed, while less iterations compromise performance noticeably.
> * Testing for $\lambda=0.5$ seems to compromise performance a lot, down to 47.9
>
> Training and testing with the same hyper-parameters:
> * For 10 iterations and
>   * $\lambda=0.01$: achieves 47.4
>   * $\lambda=1$: achieves 48.4
>   * $\lambda=0.5$: achieves 51.7, which is lower than the default model but higher than the test-time change of $\lambda=0.5$, which means that the model is sensitive to such a train-test time discrepancy.
>   * $\lambda$ is learnable:  achieves 54.0. This achieves comparable performance to the fixed value, while the final learnable value is approximately equal to 0.4.
> * For  $\lambda=0.1$
>   * 3 iterations: achieves 53.7. This is a small drop compared to the default model, but interestingly the difference is large compared to the test-time change of the number of iterations (51.9 vs 53.7). Therefore, the sensitivity to a discrepancy in the number of iterations used in training and testing is also large
>   * 5 iterations: achieves 53.6, which is a similar observation as above.
>   * 20 iterations: achieves 54.1. This is a small increase, at the cost of slower training and testing.
> * 20, 30 iterations for $\lambda=0.5$: We trained with these settings to test the hypothesis of the reviewer, based on Cuturi’s paper, that larger $\lambda$ may do better for more iterations. Both models achieve 51.8 average performance. We conclude that our settings are nearly optimal and offer a good performance-time trade-off.
>
> We will add those results in the updated manuscript.
>
> ___
> > **Extreme Domains**
>
> This is a great comment. It has been challenging for us to find instance-level datasets from extreme domains, and we would greatly appreciate any suggestions from the reviewer.
>
> Given the reviewer’s mentioning of remote sensing datasets, and despite the fact that this is not an instance-level task. We run ELVIS, trained on landmarks, on PatternNet, a remote sensing image retrieval dataset. Global descriptor (DINOv2) achieved 90.1, leaving little space for improvements, and ELViS effectively gave a small boost to 90.6, while AMES, as the second best model, reached only 90.2.
>
> To further simulate scenarios with radically different visual statistics from natural images, we will also evaluate ELViS on a stylized version of one of our existing datasets. We are currently preparing such an experiment.

---

> > ### Author Response · Authors · 2025-11-20
> >
> > > **Small-Sample Training**:
> >
> > This is an excellent suggestion, thank you. We trained ELViS on 8k images (250 instances/landmarks from GLDv2), which is roughly 1% of the training set used for the default model.
> > This model achieves 49.0, which is still a noticeable improvement compared to global descriptors (39.8).
> > This is also better than the training-free Chamfer+OT (48.2).
> >
> > Our best efforts to train an effective AMES model on the same training set result in a model that does not improve global descriptor performance. We continue our efforts with seeing how much data AMES needs to become effective and/or training other models on this small training set. We will report the updates.
> >
> > We will enrich our manuscript next week with such results.
> >
> > We would also like to emphasize that one of the main motivations behind ELViS is precisely the scenario where limited training data is available; our goal was to design a method with stronger inherent generalization. Thank you for the comment!

---

> > > ### Author Response · Authors · 2025-11-28
> > >
> > > > **Extreme Domains**
> > >
> > > We attempted to apply stylization to the existing datasets using recent diffusion models but did not manage to maintain the instance-level characteristics which violates the ground-truth of the datasets and the focus on instance-level retrieval of this work.
> > >
> > > Following the reviewer's advice to evaluate on test sets from different domains, such as remote sensing. We additionally evaluate a retrieval benchmark with satellite and astronaut photos, introduced in EarthLoc [A]. ELViS is shown highly effective compared to no reranking. Moreover, it outperforms AMES and surpasses the top results reported in the original paper. The full results are provided in the table below and will be integrated into the updated manuscript.
> > >
> > > | Method | Alps R@1 | Alps R@10 | Alps R@100 | Cal R@1 | Cal R@10 | Cal R@100 | Gobi R@1 | Gobi R@10 | Gobi R@100 | Amaz R@1 | Amaz R@10 | Amaz R@100 | Tosh R@1 | Tosh R@10 | Tosh R@100 |
> > > | :--- | :---: | :---: | :---: | :---: | :---: | :---: | :---: | :---: | :---: | :---: | :---: | :---: | :---: | :---: | :---: |
> > > | **AnyLoc** | 40.7 | 70.8 | 92.0 | 48.7 | 75.0 | 91.6 | 28.7 | 57.0 | 81.7 | 38.6 | 63.8 | 86.2 | 63.7 | 84.5 | 96.3 |
> > > | **EarthLoc** | 53.9 | 71.9 | 87.2 | 55.9 | 74.6 | 91.6 | 46.8 | 65.0 | 82.9 | 45.6 | 66.6 | 82.4 | 67.6 | 80.3 | 91.9 |
> > > | **No-rerank**| 35.7 | 65.6 | 91.1 | 50.1 | 78.2 | 93.9 | 26.2 | 53.3 | 84.0 | 31.4 | 61.1 | 83.9 | 52.4 | 78.1 | 94.9 |
> > > | **AMES** | 44.0 | 74.5 | 93.7 | 57.4 | **83.0** | **95.7** | 36.0 | 64.7 | **87.0** | 40.4 | 69.2 | 87.8 | 61.5 | 85.4 | 96.8 |
> > > | **ELViS** | **55.1** | **80.0** | **94.6** | **62.0** | 82.7 | 94.7 | **49.7** | **71.2** | 86.9 | **54.4** | **76.2** | **89.0** | **77.0** | **90.9** | **97.6** |
> > >
> > >
> > > [A] Berton et al. ”EarthLoc: Astronaut Photography Localization by Indexing Earth from Space.” CVPR, 2024.
> > >
> > > ___
> > > > **Small-Sample Training**
> > >
> > > We additionally trained ELViS on 1k, 8k, and 40k training images (corresponding to 50, 250, and 1250 GLDv2 classes), and achieved 47.9, 49.0, 51.3 mAP on average of 8 datasets, respectively. AMES was not effective, i.e. did not improve the global descriptor in any of these cases. These results will be integrated in the updated manuscript.

---

### Author Response · Authors · 2025-12-02

Dear Area Chair and Reviewers,

We have uploaded a revised version of the manuscript to address the points raised in the reviews. The changes are highlighted in red in the PDF. The updates include:

**Experiments**:
- SuperGlue comparison: Added performance comparison in Table 1.
- Small-sample training: Integrated results for training ELViS with limited training sets in Table 6 (Section 4.2).
- Improving in-domain results: Added experiment improving in-domain performance by combining ELViS and AMES in Table 5 (Section 4.2).
- Extreme domains: Added full results for the Astronaut Photography Localization benchmark in Appendix C (Table H).
- OT hyperparameters: Added an analysis of the hyperparameters in Appendix D (Table I).

**Clarifications**:
- Extended the description and motivation of function g in Section 3.4. and clarified the caption of Figure 3
- Simplified the Related Work section, moving the relevant discussion to the Introduction and adding the related work of Chowdhury et al.
- Added a summary of the proposed method at the end of Introduction.
- Expanded Appendix to include training/validation statistics (Table G), dataset visual examples (Figure E), and separate scores for Revisited Oxford and Paris (Tables J and K).

We thank the reviewers for their constructive feedback.

---

### Meta-Review · Area_Chair_UKEq · 2026-01-06

**Summary:**

The work proposes an efficient visual similarity framework built on top of strong local descriptors, with a particular focus on cross-domain generalization. The reviewers generally found the paper well-motivated and technically sound. Reviewers appreciated the simplicity and practicality of the approach, as well as the empirical evidence showing improved robustness across domains compared to heavier feature-processing pipelines. The main concerns raised were primarily about clarity, experimental reporting, and the scope of evaluation, rather than fundamental flaws in the method. Overall, the reviewer scores indicate sufficient support for acceptance.

**Reviewer Concerns:**

Concerns addressed by the rebuttal:
The rebuttal addressed several key reviewer concerns. In particular, the authors clarified the choice of optimal transport hyperparameters and added additional analysis to justify these design decisions. They also provided further experimental results in smaller training regimes, helping to demonstrate that the proposed method does not rely on extremely large training sets to generalize. In addition, the rebuttal improved clarity around dataset splits, evaluation protocols, and presentation details, which were repeatedly requested by reviewers.

Existing concerns:
Some concerns remain regarding the breadth of the cross-domain evaluation, especially on more extreme domain shifts, and the absence of direct comparisons with certain classical matching-based re-ranking pipelines. There are also minor questions about architectural design choices and presentation clarity. However, these issues primarily suggest directions for future work and do not undermine the validity or usefulness of the proposed method.

**Reviewer Scores:**

UYEL: Likely to maintain their original score, as their concerns were largely related to clarity and experimental justification, which were addressed in the rebuttal.

2TKs: Likely to maintain their score. While some questions about evaluation scope remain, the overall contribution and empirical results still support acceptance.

dvx4: Might slightly increase or maintain their score after discussion, given the added ablations and clarifications provided in the rebuttal.

6e19: Likely to maintain their original score, as the remaining concerns are minor and do not affect the core contribution.

---

### Decision · Program_Chairs · 2026-01-26

Accept (Poster)